# Selective loss of CD107a TIGIT+ memory HIV-1-specific CD8+ T cells in PLWH over a decade of ART

Oscar Blanch-Lombarte[1,2], Dan Ouchi[1], Esther Jimenez-Moyano[1], Julieta Carabelli[1], Miguel Angel Marin[1], Ruth Peña[1], Adam Pelletier[3], Aarthi Talla[3], Ashish Sharma[3], Judith Dalmau[1], José Ramón Santos[4,5], Rafick-Pierre Sékaly[3], Bonaventura Clotet[1,4,5,6,7], Julia G Prado[1,6,8]*

[1]IrsiCaixa AIDS Research Institute, Barcelona, Spain; [2]Universitat Autònoma de Barcelona, Cerdanyola del Vallès, Barcelona, Spain; [3]Pathology Department, Case Western Reserve University, Cleveland, United States; [4]Lluita contra la SIDA Foundation, Hospital Universitari Germans Trias i Pujol, Barcelona, Spain; [5]Infectious Diseases Department, Hospital Universitari Germans Trias i Pujol, Badalona, Spain; [6]Germans Trias i Pujol Research Institute (IGTP), Badalona, Spain; [7]Faculty of Medicine, University of Vic - Central University of Catalonia (UVic-UCC), Catalonia, Spain; [8]CIBER Enfermedades Infecciosas (CIBERINFEC), Instituto de Salud Carlos III, Madrid, Spain

*For correspondence:
jgarciaprado@irsicaixa.es

Competing interest: The authors declare that no competing interests exist.

**Abstract** The co-expression of inhibitory receptors (IRs) is a hallmark of CD8+ T-cell exhaustion (Tex) in people living with HIV-1 (PLWH). Understanding alterations of IRs expression in PLWH on long-term antiretroviral treatment (ART) remains elusive but is critical to overcoming CD8+ Tex and designing novel HIV-1 cure immunotherapies. To address this, we combine high-dimensional super-vised and unsupervised analysis of IRs concomitant with functional markers across the CD8+ T-cell landscape on 24 PLWH over a decade on ART. We define irreversible alterations of IRs co-expression patterns in CD8+ T cells not mitigated by ART and identify negative associations between the frequency of TIGIT+ and TIGIT+ TIM-3+ and CD4+ T-cell levels. Moreover, changes in total, SEB-activated, and HIV-1-specific CD8+ T cells delineate a complex reshaping of memory and effector-like cellular clusters on ART. Indeed, we identify a selective reduction of HIV-1 specific-CD8+ T-cell memory-like clusters sharing TIGIT expression and low CD107a that can be recovered by mAb TIGIT blockade independently of IFNγ and IL-2. Collectively, these data characterize with unprec-edented detail the patterns of IRs expression and functions across the CD8+ T-cell landscape and indicate the potential of TIGIT as a target for Tex precision immunotherapies in PLWH at all ART stages.

## Editor's evaluation

This important study shows that the expression of some inhibitory receptors on CD8 T cells is increased in people living with HIV (PLWH) and remains elevated even after years of viral suppression by antiretroviral therapy. The authors further provide convincing evidence that inhibition of TGIT partially restores the ability of CD8 T cells to produce CD107a but not the other functions and is relevant for researchers and clinicians interested in viral infections, especially HIV/AIDS.

## Introduction

The ART introduction has been the most successful strategy to control viral replication, transforming HIV-1 into a chronic condition. However, ART does not cure the infection, and treatment is required lifelong due to a stable viral reservoir, raising the need to find a cure for people living with HIV-1 (PLWH). A sterilizing or functional cure aims to eliminate or control HIV-1 in the absence of ART. In both scenarios, HIV-1-specific CD8+ T-cells are likely to play an essential role as they have been widely recognized as a critical factor in the natural control of viral replication (*Borrow et al., 1994*; *Collins et al., 2020*; *McBrien et al., 2018*; *Cartwright et al., 2016*; *Goulder and Walker, 2012*). Proliferative capacity, polyfunctionality, and ex vivo antiviral potency are features of HIV-1-specific CD8+T cells associated with spontaneous viral control (*Sáez-Cirión et al., 2007*; *Migueles et al., 2008*; *Nguyen et al., 2019*; *Perdomo-Celis et al., 2019a*).

Although ART in PLWH normalizes the levels of CD8+ T-cells and potentially preserves their functional characteristics (*Perdomo-Celis et al., 2019a*; *Perdomo-Celis et al., 2019b*; *Tavenier et al., 2015*; *Serrano-Villar et al., 2014*), microbial translocation and continuous immune activation lead to long-term CD8+ T-cell dysfunction and exhaustion (Tex), a critical barrier for HIV-1 curative interventions (*Ruiz et al., 2018*; *Jiang et al., 2020*; *Hoffmann et al., 2016*). CD8+ Tex is defined by the persistent co-expression of inhibitory receptors (IRs) and the progressive loss of immune effector functions linked to transcriptional, epigenetic and metabolic changes (*Sekine et al., 2020*; *Buggert et al., 2014*; *Bengsch et al., 2016*; *Sen et al., 2016*). In HIV-1 infection, IRs are continuously expressed despite long-term suppressive ART (*Breton et al., 2013*; *Yamamoto et al., 2011*; *Cockerham et al., 2014*; *Rutishauser et al., 2017*; *Tauriainen et al., 2017*; *Chew et al., 2016*; *Trautmann et al., 2006*) and have been associated with disease progression and immune status in PLWH (*Chew et al., 2016*; *Day et al., 2006*; *Graydon et al., 2019*; *Gupta et al., 2015*; *Jones et al., 2008*; *Teigler et al., 2017*; *Tian et al., 2015*). Thus, the expression of IRs is linked to diminished functionality and is a hallmark of CD8+ Tex in PLWH.

The interest in evaluating the blocking of IRs or immune checkpoint blockade (ICB) in PLWH as a therapeutic strategy to reverse CD8+ Tex increases (*Trautmann et al., 2006*; *Day et al., 2006*). Over the last years, several studies supported the recovery of proliferative capacity, cell survival, and cytokine production of HIV-1-specific CD8+ T-cells through ICB (*Chen et al., 2020*; *Blanch-Lombarte et al., 2019*). The blockade of the PD-1/PDL-1 axis has been extensively studied, demonstrating the functional recovery of HIV-1-specific CD8+ T-cells in PLWH (*Trautmann et al., 2006*; *Day et al., 2006*). Moreover, alternative pathways to PD-1 /PDL-1, including LAG-3, TIGIT, and TIM-3, have been explored as candidates for ICB therapies for HIV-1 infection (*Chen et al., 2020*; *Sakuishi et al., 2010*; *Harjunpää and Guillerey, 2020*; *Maruhashi et al., 2020*). Also, recent data support the combinatorial use of ICB to favour synergistic effects on the recovery of HIV-1-specific CD4+ and CD8+ T-cell function (*Attanasio and Wherry, 2017*; *Chiu et al., 2022*).

The unprecedented success of the clinical use of ICB in the cancer field (*Vaddepally et al., 2020*; *Twomey and Zhang, 2021*) has prompted the clinical evaluation of ICB in PLWH (*Gonzalez-Cao et al., 2020*) to boost immunity to reduce or eliminate the viral reservoir. However, clinical evidence on the impact of ICB as an HIV-1 cure intervention continues to be controversial (*Blanch-Lombarte et al., 2019*; *Fromentin et al., 2019*; *Le Garff et al., 2017*; *Gay et al., 2017*; *Guihot et al., 2018*; *Uldrick et al., 2022*). In this context, the simultaneous characterization of IRs co-expression and functional patterns across CD8+ T-cells is critical to understanding Tex regulation in PLWH. This information is essential to identify novel targets for precise immunotherapies in PLWH on ART (*Deeks et al., 2021*).

To address these questions, we performed supervised and unsupervised immunophenotypic analyses of IRs (PD-1, TIGIT, LAG-3, TIM-3, CD39), and functional markers (CD107a, IFNγ, and IL-2) across the landscape of CD8+ T-cells over a decade of ART in PLWH and compared to PLWH with early infection and healthy individuals. We profile changes of bulk, SEB-activated, and HIV-1-specific CD8+ T-cells in PLHW and unfold a selective decrease of memory-like HIV-1-specific CD8+ clusters sharing TIGIT expression and low CD107a in PLWH on ART. Moreover, TIGIT blockade rescues CD107a expression without changes in IFNγ or IL-2 production on HIV-1-specific CD8+ T-cells. Of note, the response to TIGIT, TIM-3, or TIGIT + TIM-3 blockade was heterogeneous across HIV-1-specific CD8+ T-cell differentiation stages and functions, indicating the plasticity and complexity of the IRs pathways as targets for immune-base cure interventions.

## Results

### Alterations in CD8+ T-cell IRs frequencies and expression patterns in PLWH are not mitigated by ART

Although the co-expression of IRs is a hallmark of CD8+ Tex in HIV-1 infection (*Breton et al., 2013*; *Yamamoto et al., 2011*; *Cockerham et al., 2014*; *Rutishauser et al., 2017*; *Tauriainen et al., 2017*; *Chew et al., 2016*; *Trautmann et al., 2006*), a detailed characterization of the combinatorial expression of IRs across CD8+ T-cell lineages in PLWH on long-term ART is still missing. To do this, we combined the analyses of IRs (PD-1, TIGIT, LAG-3, TIM-3, and CD39) and lineage markers (CD45RA, CCR7, and CD27) in CD8+ T-cells from longitudinal samples in PLWH on ART by flow-cytometry (*Figure 1—figure supplement 1*). We compare three groups; healthy controls (HC), PLWH with early infection (Ei), and PLWH on ART (S) with longitudinal samples available a median period of 2.2 (S1) and 10.1 (S2) years on fully suppressive ART (*Figure 1A*, *Figure 1—source data 1*). The epidemiological and clinical characteristics of the study groups are detailed in *Supplementary file 1* and *Figure 1—source data 1*.

As shown in *Figure 1* (*Figure 1—source data 2*), we found persistent alterations in the expression and co-expression of IRs in naïve, central memory (CM), and transitional memory (TM) CD8+ T-cells in PLWH on ART. These perturbations were maintained despite prolonged ART (S2) in naive and CM, but not TM cells compared to HC. Deconvolution of IRs co-expression patterns by the number of receptors expressed (0, 1, 2, >3) further delineates a significant reduction of Naïve, CM, and TM CD8+ T-cells lacking IRs expression concomitant with a significant increase of CD8+ T-cells expressing one (Naïve and CM) or >3 IRs (TM) on ART (*Figure 1C*). Moreover, we observed an augment in effector memory (EM), TM, and effector (EFF) CD8+ T-cells co-expressing >3 IRs in Ei and S1 that normalized in S2 (*Figure 1C*). Of note, out of the 32 possible combinations of IRs expression studied in CD8+ T-cell subsets (*Figure 1—figure supplement 2*, *Figure 1—source data 3*), single TIGIT$^+$ expression in CM and dual TIGIT$^+$TIM-3$^+$ co-expression in EFF CD8+ T-cells accounted for continuous increases in frequency under suppressive ART (*Figure 1D*, *Figure 1—source data 3*). We confirmed similar increases in the frequency of TIGIT in total, CM, and TM CD8+ T-cells on ART. These data contrast with transient changes in other IRs upon infection normalizing with ART (*Figure 1—figure supplement 2*, *Figure 1—source data 4*).

These initial findings led us to postulate associations between the expression of IRs in CD8+ T-cells, persistent immune activation, and the degree of CD4+ T-cell immune recovery in PLWH on ART. For this purpose, we performed correlation analyses that revealed several negative associations between the frequency of CD8+ T-cells expressing IRs and CD4+ T-cell counts across study groups (*Figure 1E–G*, *Figure 1—source data 5*). Focusing on S1 (*Figure 1F*), we found significant negative correlations between CD4+ T-cell counts and frequencies of CM CD8+ T-cells expressing 2 (p=0.0054, r=−0.64) and >1 IRs (p=0.0087, r=−0.61). Focusing on S2 (*Figure 1G*), we observed significant negative correlations between CD4+ T-cell counts and the frequency of total CD8+ T cells expressing TIGIT$^+$ (p=0.0157, r=−0.58), expressing 1 IRs (p=0.0386, r=−0.54) or >1 IRs (p=0.0386, r=−0.51). At the level of CD8+ T-cell subsets, the expression of 2 IRs in CM and >1 IR in TM negatively correlated with CD4+ T-cell counts (p=0.0072, r=−0.64; p=0.0346, r=−0.52, respectively; *Figure 1G*). These correlations further indicate a negative relationship between IRs expression patterns and immune status in PLWH on long-term suppressive ART.

In summary, these data support changes in IRs expression not mitigated by long-term ART in total and CD8+ T-cell subsets expressing one or >1 IRs, particularly TIGIT$^+$ and TIGIT$^+$TIM-3$^+$. These findings also uncover negative associations between IRs expression in CD8+ T-cells and CD4+ T-cell levels in PLWH on ART.

### Unsupervised phenotypic analyses of IRs across the CD8+ T-cell landscape in PLWH on ART

Next, to further characterize IRs expression across CD8+ T cells in PLWH on ART, we performed an unsupervised net-SNE analysis of flow-cytometry data. We concatenated 1,988,936 total CD8+ T-cells and analyzed the phenotypes with the topographical regions of each surface marker tested (*Figure 2A-B*, *Figure 2—source data 1*). CD8+ T cells were classified into 38 cellular clusters distributed according to the relative marker expression of 14 parameters and represented using net-SNE and heatmaps (*Figure 2C-D*).

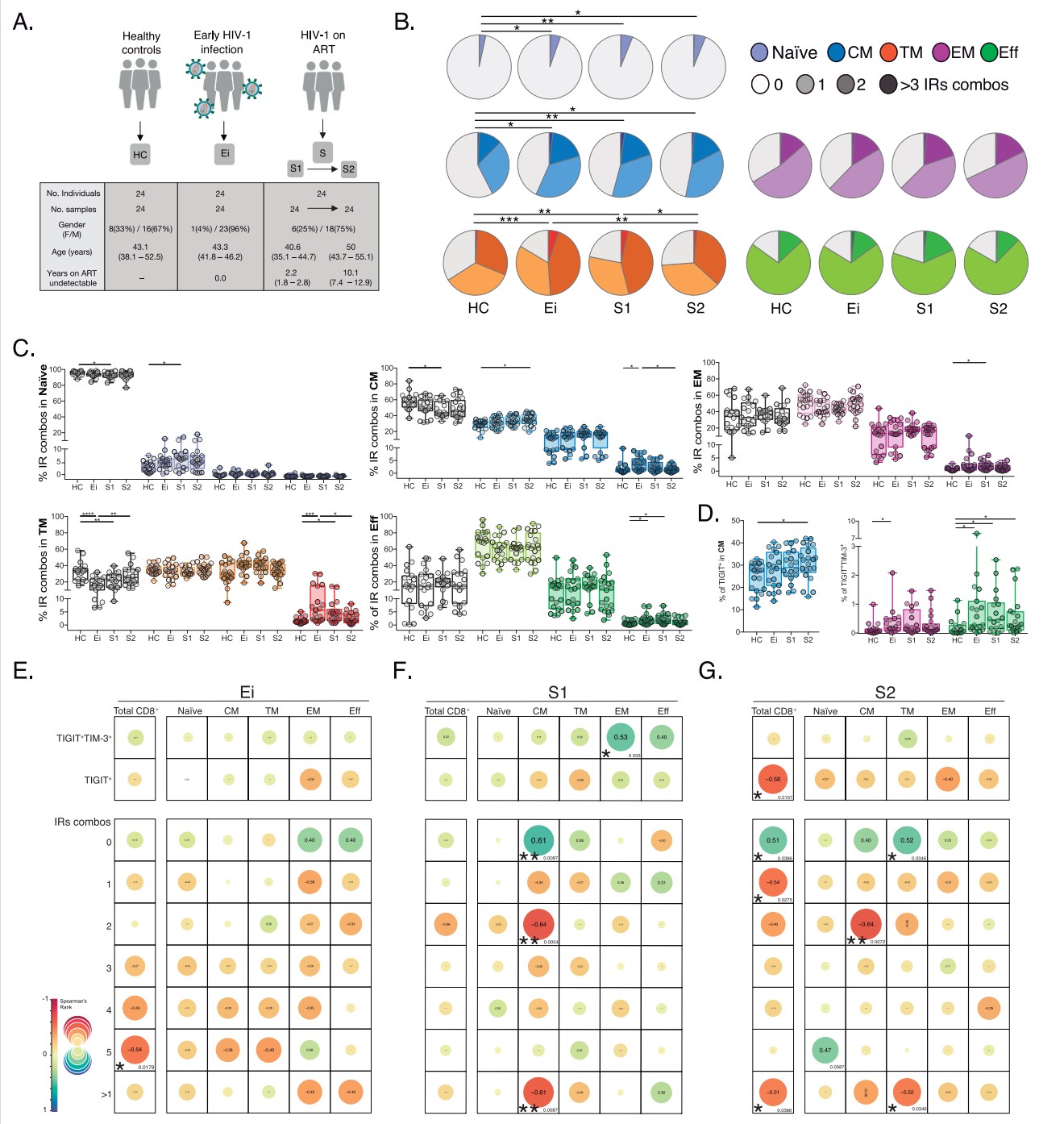

**Figure 1.** Patterns of IRs co-expression and correlations with CD4+ T-cell counts in PLWH. (**A**) Overview of study design and study groups, healthy controls (HC), PLWH in early HIV-1 infection (Ei), and PLWH on fully suppressive ART (S) in S1 and S2 time points. (**B**) The expression of IRs summarized in the pie chart is none, one, two, or more than three IRs expressed in CD8+ T-cell subsets. For statistical analysis, we used permutation tests using SPICE software. (**C**) Scatter plots showing the median and interquartile ranges of IR combinations in CD8+ T- cell subsets. (**D**) Scatter plots of the frequencies of single TIGIT+ expression in CM and TIGIT+TIM-3+ expression in EM and effector EFF CD8+ T cells. (**E–G**) Correlations between CD4+ T-cell counts as a function of TIGIT+, TIGIT+TIM-3+, and combinations of IRs from total CD8+ T-cells and subsets in Ei (**E**), S1 (**F**), and S2 (**G**). The data in B to D represent the mean of two technical replicates. We used the Mann-Whitney U test for intergroup comparison (HC, Ei, S1, and S2) and the signed-rank test for intragroup comparison (S1 and S2). Holm's method was used to adjust statistical tests for multiple comparisons. All possible correlations of the 32 Boolean IRs combinations are not shown. p-values: *<0.05, **<0.005 and ***<0.0005. Sample sizes in **A**: HC (24), Ei (24), S1(24), S2 (24). Sample sizes in **B–G**: HC (20), Ei (21), S1(18), S2 (21).

The online version of this article includes the following source data and figure supplement(s) for figure 1:

*Figure 1 continued on next page*

*Figure 1 continued*

**Source data 1.** Epidemiological and clinical data of study groups; healthy controls (HC), PLWH in early infection (Ei), and PLWH on fully suppressive ART (**S**) in S1 and S2 time points.

**Source data 2.** Frequencies of IRs expression shown in pie charts as 0 to >3 IRs expressed in CD8+ T-cell subsets and study groups.

**Source data 3.** Frequencies of the 32 possible combinations of expressions for TIGIT, PD-1, LAG-3, TIM-3, and CD39 in CD8+ T-cell subsets per study group.

**Source data 4.** Frequencies of IRs expression in total and CD8+ T-cell subsets per study group.

**Source data 5.** Correlations between CD4+ T-cell counts and TIGIT+, TIGIT+TIM-3+, and combinations of IRs in total and CD8+ T-cell subsets per study group.

**Figure supplement 1.** Gating strategy for identifying total CD8+ T-cell subsets and IRs expression levels.

**Figure supplement 2.** Expression profile of IRs in total and CD8+ T-cell subsets across study groups.

Out of the 38 clusters identified, we found eight cellular clusters (#1, #2, #6, #7, #9, #10, #11, and #12) with significant differences by inter- and intragroup comparisons (*Figure 2D–E*). Most of the differentially expressed clusters shared memory-like phenotypes. Of note, clusters #6, #7, and #12 shared memory-like phenotypes and low expression of IRs. Meanwhile, clusters #9 and #10 shared effector-like phenotypes and co-expression of IRs, including TIGIT, LAG-3, and low TIM-3 (*Figure 2D*). Briefly, intergroup analyses demonstrated significant changes in composition and frequency with a decrease of #1, #6, and an increase of #10 in Ei compared with HC. Also, clusters #1 and #2 decreased in S1 and S2, respectively, and clusters #9, #10, #11, and #12 increased in S1 while tended to normalize in S2 compared with HC (*Figure 2E*, *Figure 2—source data 2*). Intragroup analyses further supported changes in cluster frequency and composition during long-term ART. Meanwhile, clusters #6 and #7 increased, and #9 and #11 decreased over time on ART (*Figure 2F*). These data support an expansion of memory-like clusters with low IR expression (#6 and #7) and a contraction of effector-like clusters sharing TIGIT expression (#9 and #11) during ART. Thus, unsupervised analyses support cellular clusters' continuous expansion and contraction in frequency and composition across the landscape of CD8+ T-cells in PLWH on ART.

## Unsupervised phenotypic characterisation of SEB-activated CD8+ T cells in PLWH on ART

Then, we evaluate CD8+ T-cell responses by bacterial superantigen activation with Staphylococcal enterotoxin B (SEB). Using SEB can provide complementary information on T-cell activation in response to pathogens involved in the disease by stimulating TCR-VB clonotypes (*Teigler et al., 2017*; *Kou et al., 1998*; *Chiu et al., 2022*). In this context, we analysed IRs expression and functional markers using unsupervised net-SNE analysis. We defined SEB-activated CD8+ T cells by the expression of at least one functional marker (CD107a, IFNγ, IL-2) upon incubation with SEB, as previously described (*Gaffen and Liu, 2004*; *Akdis et al., 2016*; *Voskoboinik et al., 2015*; *Aktas et al., 2009*; *Bhat et al., 2017*, *Figure 3A,B*, *Figure 3—source data 1*). We concatenated 253,021 SEB-activated CD8+ T-cells and identified 29 unique clusters represented by net-SNE and heatmaps (*Figure 3C-D*). Only six of the 29 clusters showed statistical differences by inter- and intragroup comparisons (*Figure 3D*). All differential clusters shared memory-like phenotypes (#2, #3, #5, #8, and #14) except cluster #6, with effector-like phenotype and low TIGIT expression. In addition, we observed a functional exclusion of clusters expressing IL-2 (#5 and #6) and those expressing CD107a and IFNγ (#2, #3, and #8; *Figure 3D*). Intergroup comparisons identified increases in cluster #2 and a reduction of #14 and #5 with HIV-1 infection compared to HC. Additionally, ART was linked to the decrease of #5, #6, and #14 in S1 and #3 in S2 when compared with HC (*Figure 3E*, *Figure 3—source data 2*). Moreover, intragroup analyses identified an increase in clusters #6 and #14 and a reduction in #3 and #8 on ART. Of note, #6 characterizes by IL-2 expression in the without TIGIT expression. Meanwhile, clusters #3 and #8 express CD107a, IFNγ and variable expression of IRs (*Figure 3F*). In agreement with unsupervised clustering analyses, classical supervised analyses identified an augment of CD107a and IFNγ SEB-activated CD8+ T-cells with Ei that normalized over time on ART. Also, we delineate significant increases of IL-2 SEB-activated CD8+ T-cells across subsets and time on ART (*Figure 3—figure supplement 1*, *Figure 3—source data 3*).

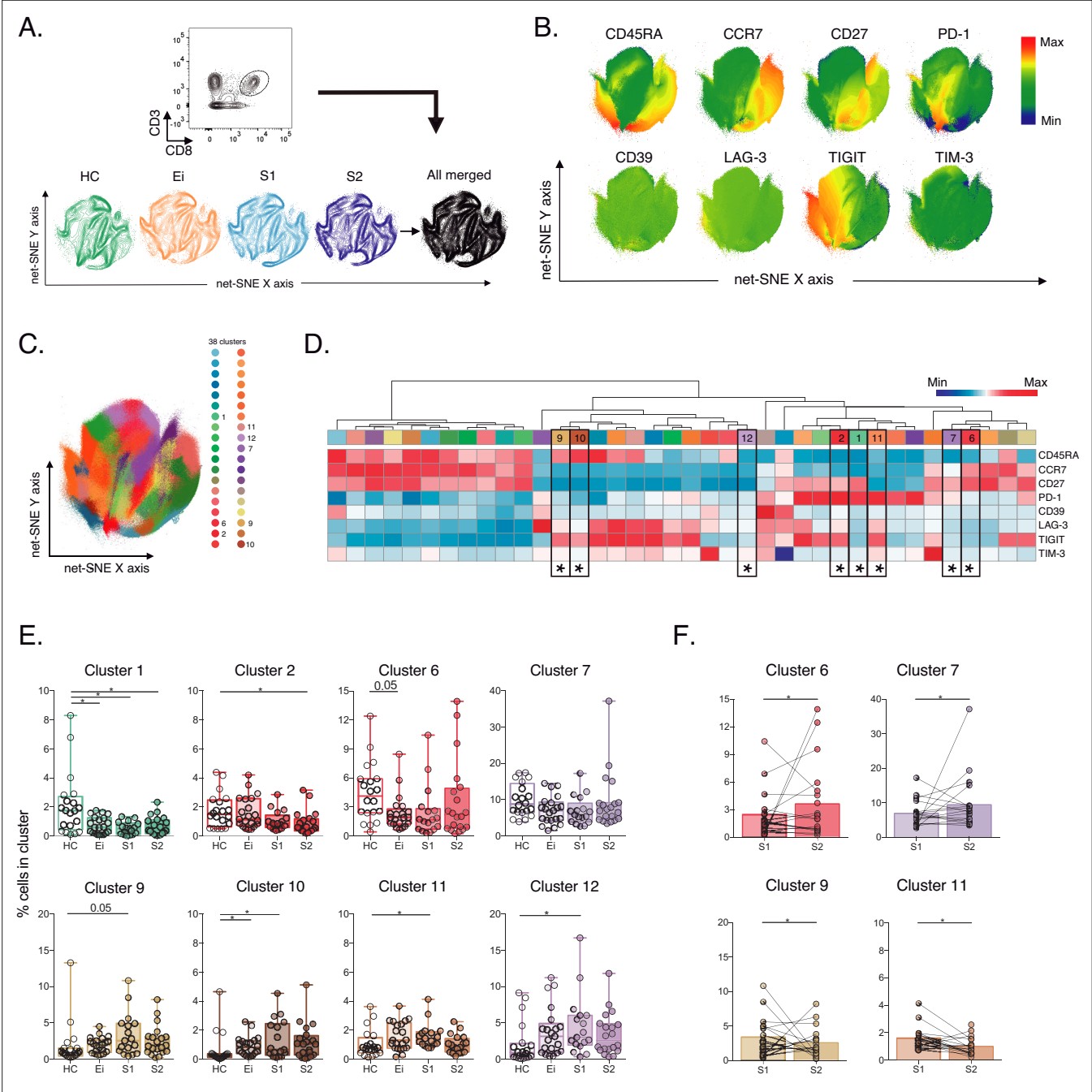

**Figure 2.** Unsupervised net-SNE analyses of total CD8+ T-cells. (**A**) Gating strategy for selecting total CD8+ T-cells (top), net-SNE plots of HC, Ei, S1, S2 and all merge groups. (**B**) Representative net-SNE visualization of surface markers. The colour gradient displays the relative marker expression. (**C**) Unsupervised KNN algorithm of 38 clusters colored according to the legend. Only clusters with statistical differences are represented in the legend. (**D**) Heatmap of the median biexponential-transformed marker expression normalized to a –3–3 range of respective markers. Asterisks represent the clusters with statistical differences. (**E–F**) Scatter plots of intergroup (HC, Ei, S1 and S2) and intragroup (S1 and S2) cluster comparisons. Data represent the median and interquartile ranges of cluster cell frequency. We used the Mann-Whitney U test for intergroup analyses and the signed-rank test for intragroup analyses. Holm's method was used to adjust statistical tests for multiple comparisons. p-values: *<0.05, **<0.005 and ***<0.0005. Sample sizes for **A–F**: HC (20), Ei (21), S1(18), S2 (21).

The online version of this article includes the following source data for figure 2:

**Source data 1.** Unsupervised net-SNE analyses of total CD8+ T-cells.

**Source data 2.** Cluster cell frequencies from net-SNE analyses in total CD8+ T-cells per study group.

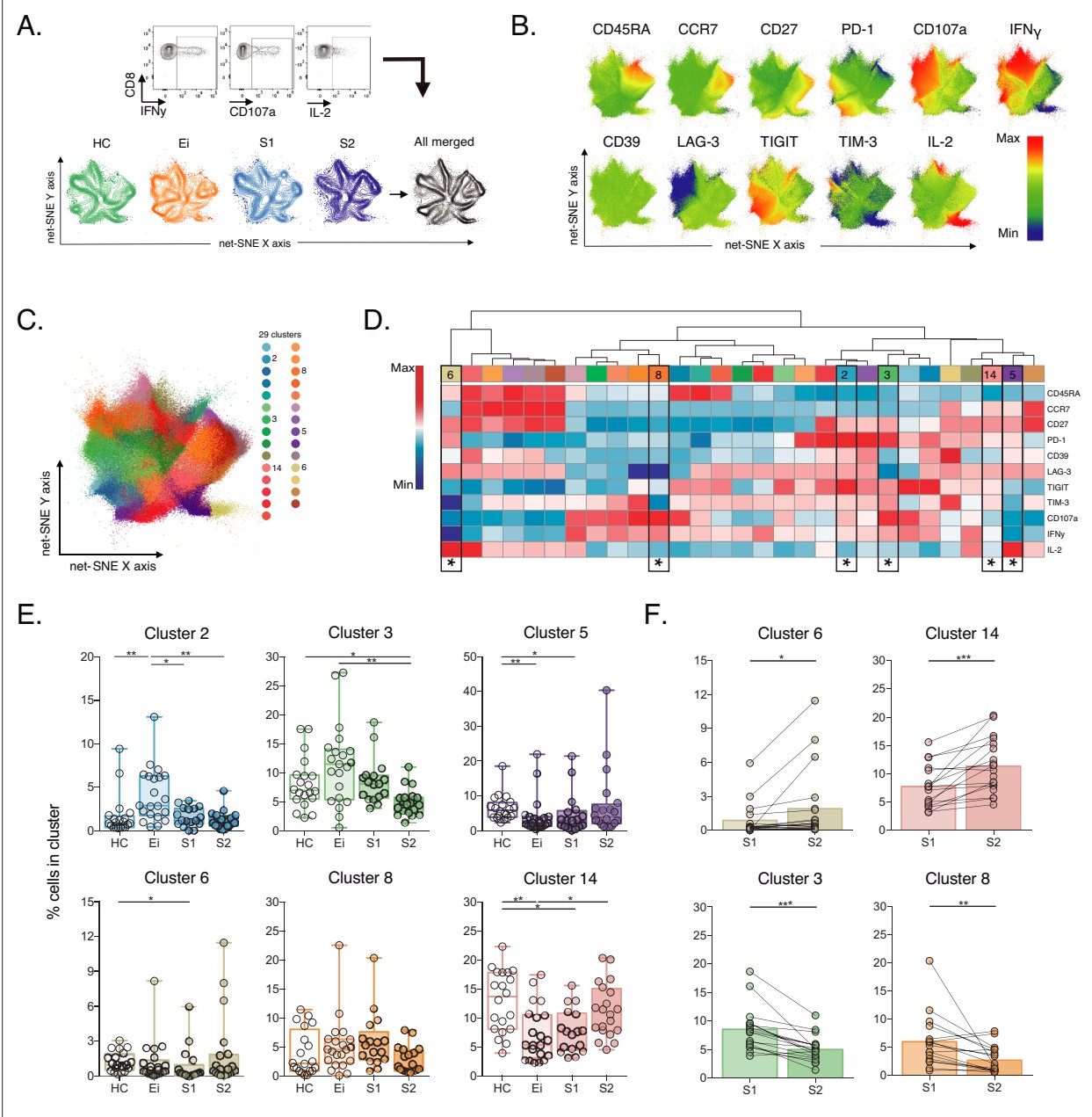

**Figure 3.** Unsupervised net-SNE analyses of SEB-activated CD8+ T-cells. (**A**) Gating representation of CD107a, IFNγ and IL-2 expression in HIV-1-specific CD8+ T-cells (top), net-SNE plots of HC, Ei, S1, S2 and merge groups of SEB-activated CD8+ T-cells (bottom). (**B**) Representative net-SNE visualization of IR expression, lineage, and functional markers. The color gradient displays relative marker expression. (**C**) Unsupervised KNN algorithm for 29 polyclonal clusters color-coded according to the legend. Clusters with statistical differences between groups are represented in the legend. (**D**) Heatmap of the median biexponential-transformed marker expression normalized to a –3–3 range of respective markers. Asterisks represent the clusters with intergroup statistical differences. (**E–F**) Scatter plots of intergroup (HC, Ei, S1 and S2) and intragroup (S1 and S2) cluster comparisons. Data represent the median and interquartile ranges of cluster cell frequency. We used the Mann-Whitney U test for intergroup analyses and the signed-rank test for intragroup analyses. Holm's method was used to adjust statistical tests for multiple comparisons. p-values: *<0.05, **<0.005, ***<0.0005. Sample sizes: HC (20), Ei (21), S1(18), S2 (21).

The online version of this article includes the following source data and figure supplement(s) for figure 3:

**Source data 1.** Unsupervised net-SNE analyses of SEB-activated CD8+ T-cells.

**Source data 2.** Cluster cell frequencies from net-SNE analyses of SEB-activated CD8+ T-cells per study group.

**Source data 3.** Supervised analyses of CD107a, IFNγ and IL-2 frequencies of SEB-activated CD8+ T-cells per study groups.

**Figure supplement 1.** Supervised analyses of SEB-activated CD8+ T-cells.

These data support plasticity in the composition of SEB-activated CD8+ T-cell clusters with HIV-1 infection and ART. We observed the dominance of memory-like clusters with changes in composition and frequency in PLWH on ART, particularly IL-2 expression.

## Reduction of HIV-1-specific CD8+ T-cell clusters sharing memory-like phenotypes, TIGIT expression and low CD107a

Next, we characterize HIV-1-specific CD8+ T-cells in PLWH on long-term ART compared to Early infected individuals (Ei), aiming to identify signatures of cellular dysfunction. We performed unsupervised net-SNE analyses in 53,751 cells concatenated, based on the production of at least one functional marker (CD107a, IFNγ, IL-2) in response to HIV-1-Gag peptides combined with lineage markers and IRs (*Figure 4A-B*, *Figure 4—source data 1*). We defined 26 HIV-1-specific clusters by net-SNE analysis. Of note, only three showed significant differences between groups (#1, #2, and #3; *Figure 4C-D*). All three clusters decreased in frequency on ART and shared memory-like features (*Figure 4E*, *Figure 4—source data 2*). Clusters #1 and #2 shared co-expression of IRs, mainly PD-1 and TIGIT, low CD107a, IFNγ, and higher expression of IL-2 than #3. Meanwhile, cluster #3 co-express (TIGIT, PD-1, and TIM-3) low expression of IL-2 and higher expression of CD107a and IFNγ (*Figure 4D*). Although CD107a, IFNγ, and IL-2 expression differed between clusters #1, #2, and #3, all shared TIGIT expression in the context of variable levels of TIM-3. These findings in memory-like clusters, together with the initial ones, accounting for increases in the frequency of TIGIT+ and TIGIT+-TIM-3+ in CM and EFF cells on ART, led us to postulate then as potential markers of HIV-1-specific CD8+ T-cell dysfunction in PLWH on ART.

Despite no significant changes observed in the total frequency of CD107a, IFNγ and IL-2 HIV-1-specific CD8+ T-cell responses between groups (*Figure 4—figure supplement 1A–B*, *Figure 4—source data 3*). The analyses of TIGIT and TIGIT +TIM-3 HIV-1-specific memory subsets revealed decreases in the frequency of CD107a TIGIT HIV-1-specific CD8+ T-cells limited to the CM compartment (*Figure 4F*, *Figure 4—source data 4*). No changes in IFNγ and IL-2 were observed. Furthermore, polyfunctional analysis of TIGIT HIV-1-specific CM CD8+ s identified a decrease in monofunctional CD107a+ as well as in bifunctional CD107a+IFNγ+ and CD107+IL-2+ cells overtime on ART (*Figure 4G*). Overall, these data support a reduction of HIV-1-specific CD8+ T-cell clusters sharing memory-like phenotypes, TIGIT expression and low CD107a in PLWH on ART.

## TIGIT blockade restores CD107a expression but not IFNγ or IL-2 production in HIV-1-specific CD8+ T cells

Then, we decided to explore the blockade of TIGIT and TIM-3 pathways as targets for the recovery of CD107a and potentially IFNγ and IL-2 production in HIV-1-specific CD8+ T-cells. We performed short-term ICB experiments using monoclonal antibodies αTIGIT, αTIM-3, and αTIGIT+ αTIM-3 in PBMC from PLWH on suppressive ART (S) with previous immunophenotype.

After short-term ICB, we monitored changes in CD107, IFNγ, and IL-2 in total and subsets of HIV-1-specific CD8+ T-cells by flow cytometry (*Figure 5A*). The Net-SNE TIGIT and TIM-3 projections are represented in *Figure 5B* (*Figure 5—source data 1*). At the level of total HIV-1-specific CD8+ T-cells, short-term ICB experiments demonstrate a specific increase of CD107a expression by αTIGIT (isotype vs. αTIGIT; p<0.05) and αTIGIT + αTIM-3 (isotype vs. αTIGIT + αTIM-3; p<0.05) blockade (*Figure 5B*, left). Moreover, the recovery of CD107a by αTIGIT was consistent across HIV-1-specific CD8+ T-cell subsets, being particularly marked for CM (isotype vs. αTIGIT; in CM p<0.005; *Figure 5C*) in agreement with our previous findings. Of note, αTIM-3 blockade did not show any effect, and dual blockade of αTIM-3+αTIGIT did not reveal an additive effect (*Figure 5C*). No changes in IFNγ and IL-2 production were observed for any conditions tested (*Figure 5D–E*, *Figure 5—source data 2*).

These data support heterogeneity in the functional recovery of HIV-1-specific CD8+ T-cells by differentiation stage based on αTIGIT, αTIM-3, and αTIGIT+αTIM-3 blockade. Overall, these results identify the targeting of TIGIT to recover the degranulation in HIV-1-specific CD8+ T-cells, particularly within the CM compartment in PLWH on ART.

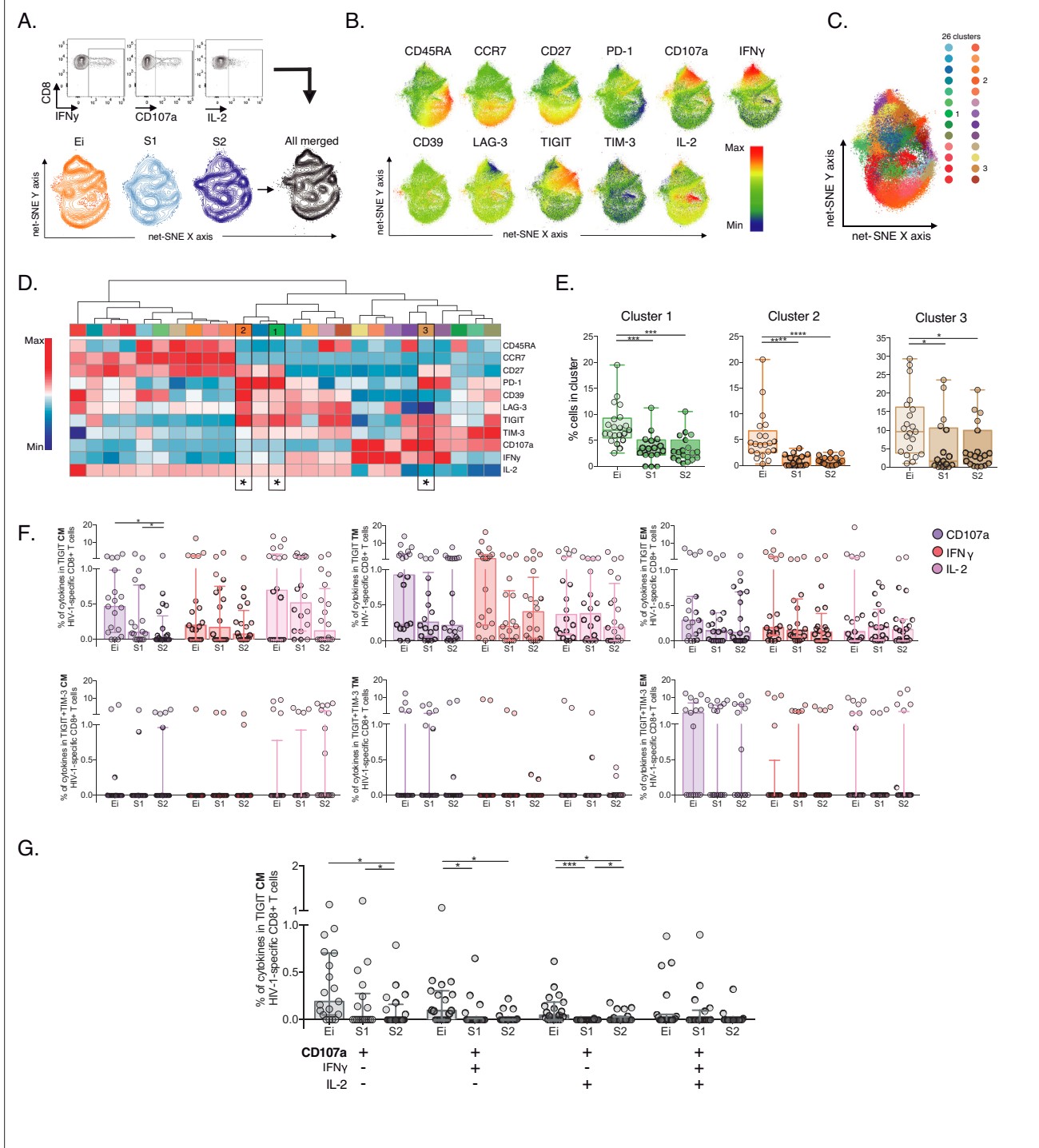

**Figure 4.** Unsupervised and supervised analyses of HIV-1-specific CD8+ T-cells. (**A**) Gating representation of CD107a, IFNγ, and IL-2 expression in HIV-1-specific CD8+ T-cells (top), net-SNE plots of Ei, S1, S2 and merge groups for HIV-1-specific CD8+ T-cells (bottom). (**B**) Representative net-SNE plots for surface and functional markers. The color gradient displays relative marker expression. (**C**) Unsupervised KNN algorithm for 26 HIV-1-specific clusters color-coded according to the legend. Only clusters with statistical differences are represented in the legend. (**D**) Heatmap of the median biexponential-transformed marker expression normalized to a −3–3 range of respective markers. Asterisks represent the clusters with intergroup statistical differences. (**E**) Scatter plots of intergroup (Ei, S1 and S2) cluster comparisons with significant statistical differences. Data represent the median and interquartile ranges of cluster cell frequency. (**F**) CD107a, IFNγ, and IL-2 frequency of expression in TIGIT+ (upper panel) and TIGIT +TIM-3+ (bottom panel) HIV-1-specific memory CD8+ T-cell subsets. Scatter plots represent the median and interquartile ranges. (**G**) Polyfunctional analyses of CD107a, IFNγ, and IL-2 expression in CM TIGIT HIV-1-specific CD8+ T-cells. Scatter plots represent median and interquartile ranges. We used the Mann-Whitney U test

*Figure 4 continued on next page*

*Figure 4 continued*

for intergroup analyses and the signed-rank test for intragroup analyses. Holm's method was used to adjust statistical tests for multiple comparisons. p-values: *<0.05, ***<0.0005, and ****<0.0001. Sample sizes: Ei (21), S1(18), S2 (21).

The online version of this article includes the following source data and figure supplement(s) for figure 4:

**Source data 1.** Unsupervised net-SNE analyses of HIV-1-specific CD8+ T-cells.

**Source data 2.** Cluster cell frequencies from net-SNE analyses of HIV-1-specific Tcells per Ei and S (**S1–S2**) study groups.

**Source data 3.** Supervised analyses of CD107a, IFNγ and IL-2 frequencies of HIV-1-specific CD8+ T-cells in Ei and S (**S1–S2**) study groups.

**Source data 4.** Frequencies of CD107a, IFNγ, and IL-2 expression in TIGIT+ and TIGIT+ TIM-3+HIV-1-specific memory CD8+ T-cell subsets in Ei and S (**S1–S2**) study groups.

**Figure supplement 1.** Supervised analyses of HIV-1-specific CD8+ T-cell responses.

## Discussion

CD8+ Tex displays a range of functional defects in PLWH early in HIV-1 infection and during ART (*Sekine et al., 2020*; *Buggert et al., 2014*; *Bengsch et al., 2016*; *Sen et al., 2016*). The expression of IRs is a hallmark of Tex (*Breton et al., 2013*; *Cockerham et al., 2014*; *Rutishauser et al., 2017*; *Jiang et al., 2020*), and co-expression of IRs has been associated with HIV-1 disease progression (*Hoffmann et al., 2016*; *Chew et al., 2016*; *Gupta et al., 2015*; *Jones et al., 2008*; *Tian et al., 2015*) and cancer severity (*Chauvin et al., 2015*; *Anderson et al., 2016*; *Joller and Kuchroo, 2017*). Although ICB has demonstrated promising results in cancer remission (*Vaddepally et al., 2020*; *Twomey and Zhang, 2021*), its applicability in HIV-1 as a cure intervention remains unclear (*Blanch-Lombarte et al., 2019*; *Fromentin et al., 2019*; *Le Garff et al., 2017*; *Gay et al., 2017*; *Guihot et al., 2018*; *Uldrick et al., 2022*). Therefore, it is essential to understand the patterns of IRs expression and function across the CD8+ T-cell landscape to identify targets for ICB broadly applicable to PLWH on ART (*Deeks et al., 2021*).

Here, we immunophenotype CD8+ T-cells using five IRs and three functional markers in PLWH over the ten years of ART. In this way, we overcome previous study limitations based on single IRs expression, bulk CD8+ T-cells, and cross-sectional data (*Breton et al., 2013*; *Rutishauser et al., 2017*; *Chew et al., 2016*; *Gupta et al., 2015*; *Tian et al., 2015*). Our results demonstrated a marked and significant increase in TIGIT CD8+ T-cells, particularly within the central memory compartment, not ameliorated by long-term ART. Also, TIGIT CD8+ T-cells negatively correlated with CD4+ counts in PLWH on ART. These data support continuous expression of TIGIT despite ART in agreement with previous studies (*Tauriainen et al., 2017*; *Chew et al., 2016*; *Jones et al., 2008*; *Anderson et al., 2016*; *Schildberg et al., 2016*) and uncover novel associations between TIGIT expression in CD8+ T-cells and poorer immune status in PLWH on ART. Thus, these data indicate a specific contribution of TIGIT expression to persistent immune activation and poor CD4+ recovery on ART.

In contrast, we observed transient increases of PD-1, LAG-3, TIM-3, and CD39 expression in total, memory and effector CD8+ T-cells that normalize over time on ART. The potential biological implications of such a difference may relate to the nature of each receptor and the specific immune regulatory pathway activated during HIV-1 infection and ART. Also, these divergences may be influenced by the presence of γ-chain cytokines, such us IL-2, IL-15, and IL-21, in plasma able to upregulate the TIGIT expression in CD8+ T-cells (*Chew et al., 2016*). A previous study demonstrated an association between high levels of IL-15 and high TIGIT expression in CD4+ T-cells with suboptimal CD4+ recovery in PLWH on ART (*Pino et al., 2021*).

Our study combined high-dimensional supervised and unsupervised analysis providing an unprecedented deep immunophenotype of the CD8+ T-cell landscape in PLWH over a decade on ART. We explored complementary levels of complexity for the characterization of CD8+ T-cells, including an absence of stimuli (bulk), the presence of antigen-independent stimuli (SEB), and the presence of antigen-specific stimuli (HIV-1).

In the absence of stimuli, supervised analyses confirmed heterogeneous and complex patterns of IRs co-expression across CD8+ T-cell lineages altered by HIV-1 infection and shaped by ART (*Sekine et al., 2020*; *Yamamoto et al., 2011*; *Jones et al., 2008*; *Noyan et al., 2018*; *Avery et al., 2018*). Furthermore, unsupervised analyses added complexity to previous data by delineating in profound detail early and continuous changes of contraction and expansion of cellular clusters with HIV-1 infection and time on ART. We tracked memory-like expansion and effector-like cluster contraction

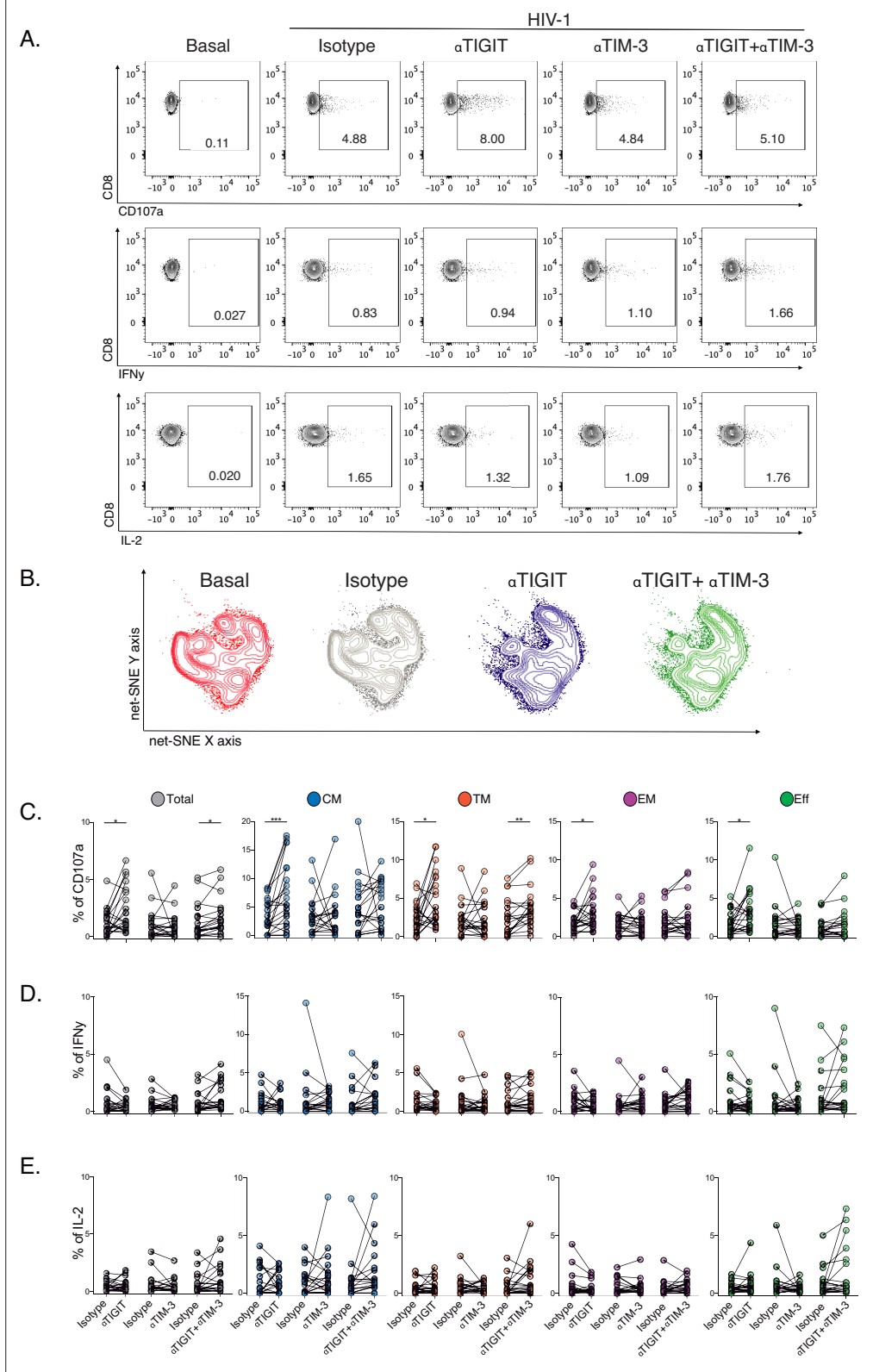

**Figure 5.** Effect of TIGIT, TIM-3, and TIGIT +TIM-3 mAb blockade in HIV-1-specific CD8+ T-cell responses in PLWH on ART. (**A**) Representative flow cytometry plots gated on CD8+ T-cells, in the absence of HIV-1 Gag stimulation (basal condition) and presence of HIV-1 Gag stimulation with isotype control, αTIGIT, αTIM-3, and αTIGIT+αTIM-3 antibodies for CD107a, IFNγ and IL-2 expression. (**B**) Representative net-SNE plots for HIV-1-specific CD8+

*Figure 5 continued on next page*

*Figure 5 continued*

T-cells from PLWH concatenated and merged according to the condition. (**C–E**) Frequency of CD107a, IFNγ, and IL-2 expression in total and HIV-1-specific CD8+ T-cell subsets for the various conditions tested. The Wilcoxon matched-pairs signed ranked test calculated statistical differences. The data represent the mean of two technical replicates. p-values:=0.05, *<0.05, **<0.005 and ***<0.0005. Sample sizes: S1(10), S2 (10).

The online version of this article includes the following source data for figure 5:

**Source data 1.** Unsupervised net-SNE analyses for HIV-1-specific CD8+ T-cells in PLWH on ART.

**Source data 2.** Frequencies of CD107a, IFNγ, and IL-2 expression in total and subsets of HIV-1-specific CD8+ T-cells in PLWH on ART.

over time on ART according to the establishment of memory responses. Similarly, in the presence of antigen-independent stimuli, we identified continuous changes in cellular cluster composition by contraction and expansion of CD8+ cellular cluster with infection and treatment. We observed a dominance of memory-like clusters with changes in composition and frequency tracked by an augment of IL-2 expressing clusters on ART both by supervised and unsupervised analyses. The IL-2 expression regulates proliferation and homeostasis and contributes to the generation of long-term memory responses (*Teigler et al., 2017*; *Kou et al., 1998*; *Chiu et al., 2022*), suggesting a partial functional remodelling of CD8+ T-cell populations independent of antigen in PLWH over time on ART. Thus, our findings support the contribution of IRs co-expression in CD8+ Tex and T-cell activation favouring a continuous reshaping of memory and effector-like CD8+ cellular clusters. These findings indicate the enormous plasticity and constant homeostasis of CD8+ T-cells in PLWH during a decade of ART (*Warren et al., 2019*).

In the presence of HIV-1-specific stimuli, supervised and unsupervised analyses delineate a reduction of HIV-1-specific CD8+ T cells sharing memory phenotypes, TIGIT expression and low CD107a. Our findings focused on the memory compartment of TIGIT expressing HIV-1-specific CD8+ T-cells demonstrating a decrease in monofunctional CD107a$^+$ and bifunctional CD107a$^+$IFNγ$^+$ and CD107a$^+$IL-2$^+$ cells. Previous studies support a direct correlation between monofunctional CD107a$^+$ and Eomes intensity in exhausted HIV-1-specific CD8+ T-cells (*Buggert et al., 2014*). Although this study did not include the expression of TIGIT across analyses, it may suggest an association between TIGIT expression and the T-bet /Eomes axis in charge of regulating the exhaustion and memory cell fate of CD8+ T-cells (*Doering et al., 2012*).

Our findings further support the role of TIGIT as a signature of dysfunctional and Tex antiviral responses (*Chew et al., 2016*). Indeed, the blockade of the TIGIT pathway restored CD107a expression in HIV-1-specific CD8+ T-cells across cellular compartments with a marked effect in the central memory compartment according to our findings from HIV-1-specific immunophenotype of TIGIT CD8+ T-cells. To our knowledge, our study is the first to demonstrate the recovery of CD107a expression in HIV-1-specific CD8+ T-cells by TIGIT blockade. These data contrast with Chew et al., which observed the recovery of IFNγ production by TIGIT blockade. However, they also reported a reduction in CD107a expression in TIGIT+ CD8 T-cells in response to aCD3/aCD28 activation, supporting a dysfunctional profile of TIGIT+ CD8+ T-cells. Differences between study groups accounting for time on ART, samples tested and interindividual variability of in vitro ICB experiments may account for some of the differences observed.

Although the mechanistic behind TIGIT signalling and CD107a expression are not fully understood, low CD107a expression has been linked to the terminal T-bet$^{dim}$Eomes$^{hi}$ exhausted phenotype, and HIV-1-specific CD8+ T-cells expressing TIGIT can degranulate to a certain extent (*Buggert et al., 2014*; *Tauriainen et al., 2017*). Moreover, our data did not support an additive effect recovering HIV-1-specific CD8+ T-cells function of TIGIT and TIM-3 combinatorial blockade over TIGIT blockade. The redundancy and promiscuity of TIM-3 for several ligands, including Gal-9, CEACAM-1, PtdSer, and HMGB-1 (*Andrews et al., 2019*; *Sabatos-Peyton et al., 2018*), may be associated with these results. We cannot exclude the impact of TIGIT and TIM-3 blockade in other cell types (*Anderson et al., 2016*; *Joller and Kuchroo, 2017*; *Pende et al., 2006*).

We acknowledge several study limitations; First, the sample size of study groups and the use of peripheral blood samples underestimate the potential contribution of TIGIT expression to Tex in lymphoid tissues. Second, the use of only Gag as stimuli for the characterization of HV-1-specific CD8+ T- cell responses in the absence of TCR sequencing. Using alternative HIV-1 antigens such as Nef, Env

or Pol may provide additional information on the profile of CD8+ T-cell functional responses against early and late-expressed viral proteins in PLWH on ART (*Kløverpris et al., 2013*; *Stevenson et al., 2021*). Third, limited ICB experiments to CD107a, INFγ and IL-2 functional markers without complementary cytotoxic markers (perforin, granzyme B). Forth, complementary transcriptomic, epigenetic, and metabolic markers are needed for a complete description of Tex's immune signatures linked to TIGIT expression in HIV-1 specific CD8+ T-cells in PLWH on ART.

In summary, our study profile with unprecedented detail continuous reshaping of memory-like and effector-like of CD8+ T cellular clusters in PLWH over a decade of ART. The study identifies the TIGIT as a critical target for Tex associated with the loss of CD107a expression in HIV-1-specific CD8+ T-cells. These findings support targeting the TIGIT/CD155 axis for Tex precision immune-base curative interventions in PLWH at all ART stages.

# Materials and methods
## Study groups
This retrospective study analyzed clinical data and biological sample availability from 3000 patients assigned to the HIV-1 clinical unit of the Germans Trias i Pujol University Hospital. We included individuals with cryopreserved PBMCs available in our collection. We identified 24 chronically HIV-1-infected individuals who had been treated mainly with a combination of NNRTI and NRTI for more than ten years with sustained virological suppression (<50 HIV-1-RNA copies/ml) and with longitudinal biological samples at timepoint 1 (S1), 2.2 (1.8–2.8) years undetectable on ART, and at time point 2 (S2), 10.1 (7.4–12.9) years undetectable on ART (*Supplementary file 1*, *Figure 1A*, Source data *Figure 1A*). We excluded individuals with integrase inhibitors, ART as monotherapy, and treatments with mitochondrial toxicity, including Trizivir, d4T, ddI, AZT and blips over the ART period (S1- S2) to ensure homogeneous treatment over time. For comparative purposes, we included 24 early HIV-1-infected individuals (Ei) defined in a window of 1.3 (0.77–17.8) weeks after seroconversion in the absence of ART and 24 healthy controls (HC). The groups were balanced by age to the S2 samples to avoid confounding effects on IR expression.

## CD8+ T-cell immunophenotype
Cryopreserved PBMCs from the study groups were thawed and rested overnight at 37 °C in a 5% $CO_2$ incubator. The following day, PBMCs were incubated for six hours at 37 °C in a 5% $CO_2$ incubator under RPMI complemented medium 10% FBS in the presence of CD28/49d co-stimulatory molecules (1 µl/ml, BD), Monensin A (1 µl/ml, BD Golgi STOP), and anti-human antibody for CD107a (PE-Cy5, clone H4A3, Thermo Fisher Scientific). PBMCs were left unstimulated, stimulated with SEB (1 µg/ml, Sigma-Aldrich), and stimulated with HIV-1-Gag peptide pool (2 µg/peptide/ml, EzBiolab). After six hours of stimulation, cells were rested overnight at 4 °C as previously described (*Blanch-Lombarte et al., 2019*). The next day, PBMCs were washed with PBS 1 X and stained for 25 min with the Live/Dead probe (APC-Cy7, Thermo Fisher Scientific) at RT to discriminate dead cells. Cells were washed with PBS 1 X and surface stained with antibodies for 25 minutes at RT. We used CD3 (A700, clone UCHT1, BD), CD4 (APC-Cy7, clone SK3, BD), CD8 (V500, clone RPA-T8, BD), CD45RA (BV786, clone HI100, BD), CCR7 (PE-CF594, clone 150503, BD), CD27 (BV605, clone L128, BD), TIGIT (PE-Cy7, clone MBSA43, Labclinics SA), PD-1 (BV421, clone EH12.1, BD), LAG-3 (PE, clone T47-530, BD), TIM-3 (A647, clone 7D3, BD) and CD39 (FITC, clone TU66, BD) antibodies. Afterwards, cells were washed twice in PBS 1 X, fixed, and permeabilized with Fix/Perm kit (A and B solutions, Thermo Fisher Scientific) for intracellular cytokine staining with anti-human antibodies of IFNγ (BV711, clone B27, BD) and IL-2 (BV650, clone MQ1-17H12, BD). Finally, stained cells were washed twice with PBS 1 X and fixed in formaldehyde 1%.

## TIGIT, TIM-3 and TIGIT+TIM-3 short-term checkpoint blockade
We selected cryopreserved PBMCs from S1 (n=10) and S2 (n=10). Samples were previously characterized by the expression of TIGIT and TIM-3 on total CD8+ T-cells. PBMCs were thawed and rested for four hours at 37 °C in a 5% $CO_2$ incubator. Next, cells were incubated under RPMI complemented medium 10% FBS with 1 µl/ml of anti-CD28/CD49d and 1 µl/ml of Monensin A overnight at 37 °C in a 5% $CO_2$. PBMCs are divided in the following conditions; (1) unstimulated, (2) SEB (1 µg/

ml, Sigma-Aldrich) and (3) HIV-1-Gag peptide pool (2 μg/peptide/ml) in the absence or presence of αTIGIT and/or αTIM-3, and its respective isotype antibodies. For the single blockade of TIGIT (αTIGIT), we included Ultra-LEAF purified anti-human TIGIT antibody (10 μg/ml, clone A15153G, Biolegend) or its control isotype Ultra-LEAF purified mouse IgG2a antibody (10 μg/ml, MOPC-173, Biolegend). For single TIM-3 blockade (αTIM-3), we used Ultra-LEAF purified anti-human TIM-3 antibody (10 μg/ml, clone F38-2E2, Biolegend) or its respective isotype Ultra-LEAF purified mouse IgG1 antibody (10 μg/ml, MOPC-21, Biolegend). Finally, we included αTIGIT+αTIM-3 or their respective IgG2 + IgG1 isotypes for a combinational blockade. The next day, PBMCs were surface and intracellularly stained with the panel of antibodies and the methodology described in the section above.

## Supervised immunophenotype data analysis

Stained PBMCs were acquired on an LSR Fortessa cytometer using FACSDiVa software (BD). Approximately 1,000,000 events of PBMCs were recorded per specimen. Antibody capture beads (BD) were used for single-stain compensation controls. Flow cytometry data were analyzed with FlowJo software v10.6.1, and fluorescence minus one (FMO) was used to set manual gates. We analyzed CD8+ T-cells by excluding dump and CD4+ T-cells. We excluded patients with <20% viability in lymphocytes and total CD8+ T-cells (Source data *Figure 1A*). As previously described, we measured by supervised and classical analyses the IRs expression in CD8+ T-cell subsets, including Naive, central memory, transitional memory, effector memory, and effector CD8+ T-cells (*Breton et al., 2013*; *Blanch-Lombarte et al., 2019*). We performed two technical replicates for SEB-activated and HIV-1-specific CD8+ T-cell cytokine production. We considered the cytokine response positive after background subtraction (mean of two technical replicates) used as the cut-off value. For each independent sample, we recorded a median of 1,000 events and 50 events positive for cytokines for total and CD8+ T-cell subsets, respectively.

## Unsupervised immunophenotype data analysis

The phenotypic and functional characterization of cellular populations was analyzed by using t-Distributed Stochastic Neighbor Embedding (t-SNE; *van der Maaten, 2008*) and net-SNE (*Cho et al., 2018*) dimensionality reduction algorithms to visualize single-cell distributions in two-dimensional maps. Briefly, cell intensity was z-normalized, and a randomly selected subset of cells, at least 1000 cells per sample, was passed through the t-SNE algorithm. The resulting t-SNE dimension was then used to predict the position of all remaining CD8+ T-cells acquired per sample from each group using the net-SNE algorithm based on neural networks. For functional analysis, we selected polyclonally activated and HIV-1-specific CD8+ T-cells producing at least one cytokine CD107a, IFNγ, or IL-2 under SEB or HIV-1 conditions, respectively. In parallel, we discovered cell communities using the Phenograph clustering technique. It operates by computing the Jaccard coefficient between nearest neighbours, which was set to 30 in all executions, and then locating cell communities (or clusters) using the Louvain method. The method creates a network indicating phenotypic similarities between cells. The netSNE maps included representations of the identified cell communities, and additionally, we built a heatmap with the clusters in the columns and the markers of interest in the rows to better comprehend the phenotypical interpretation of each cluster. The color scale displays each marker's median intensity on a biexponential scale. We calculated quantitative assessments of cellular clusters in the percentage of cells for each sample to analyze and compare the distribution between HC, Ei, S1, and S2 groups, similar to the classical flow cytometry analysis.

## Statistics

Bivariate analysis was conducted using nonparametric methods as follows: Mann-Whitney U test for independent median comparison between groups, Wilcoxon signed-rank test for paired median changes over time, permutation test for composition distribution between groups, Kruskal-Wallis test for comparison between more than two groups, and spearman linear correlation coefficient to study the association between continuous variables. Holm's method was used when appropriate to adjust statistical tests for multiple comparisons with a significance level of 0.05. All statistics and single-cell analyses were conducted using the R statistical package (*R Development Core Team, 2008*). The selection of clusters was performed through the significant differences obtained by inter- and intra-group comparisons between groups. Moreover, pattern distribution and graphical representations

of all possible Boolean combinations for IRs co-expression and functional markers were conducted using the data analysis program Pestle v2.0 and SPICE v6.0 software (*Roederer et al., 2011*). Graph plotting was performed by GraphPad Prism v8.0 software and R packages. The data represents the mean of two technical replicates.

## Study approval

The study was conducted according to the principles of the Declaration of Helsinki (*World Medical Association, 2013*). The Hospital Germans Trias i Pujol Ethics Committee approved all experimental protocols (PI14-084). For the study, subjects provided written informed consent for research purposes of biological samples taken from them.

## Acknowledgements

We thank the Flow Cytometry Core Facility from the Germans Trias i Pujol Research Institute (IGTP).

## Additional information

### Funding

| Funder | Grant reference number | Author |
|---|---|---|
| Institute of Health Carlos III | PI17/00164 | Julia G Prado |
| Catalan Government and the European Social Fund | AGAUR-FI_B 00582 PhD fellowship | Oscar Blanch-Lombarte |
| Redes Temáticas de Investigación en SIDA | ISCIII RETIC RD16/0025/0041 | Esther Jimenez-Moyano |
| Grifols | | Julia G Prado |

The funders had no role in study design, data collection and interpretation, or the decision to submit the work for publication.

### Author contributions

Oscar Blanch-Lombarte, Conceptualization, Data curation, Formal analysis, Supervision, Validation, Investigation, Methodology, Writing – original draft, Writing – review and editing; Dan Ouchi, Data curation, Software, Formal analysis, Validation, Methodology; Esther Jimenez-Moyano, Data curation, Formal analysis, Methodology; Julieta Carabelli, Miguel Angel Marin, Formal analysis, Methodology; Ruth Peña, Methodology; Adam Pelletier, Aarthi Talla, Ashish Sharma, Rafick-Pierre Sékaly, Software, Methodology; Judith Dalmau, Funding acquisition, Investigation, Project administration; José Ramón Santos, Recruited the study participants; Bonaventura Clotet, Recruited the study participants; Julia G Prado, Conceptualization, Data curation, Formal analysis, Supervision, Funding acquisition, Validation, Investigation, Methodology, Writing – original draft, Project administration, Writing – review and editing

### Author ORCIDs

Oscar Blanch-Lombarte ⓘ http://orcid.org/0000-0002-8317-7535
Miguel Angel Marin ⓘ http://orcid.org/0000-0001-5294-6007
Julia G Prado ⓘ https://orcid.org/0000-0002-5439-4645

### Ethics

Human subjects: The study was conducted according to the principles expressed in the Declaration of Helsinki (Fortaleza, 2013). The Hospital Germans Trias i Pujol Ethics Committee approved all experimental protocols (PI14-084). For the study, subjects provided their written informed consent for research purposes of biological samples taken from them.

### Decision letter and Author response

Decision letter https://doi.org/10.7554/eLife.83737.sa1
Author response https://doi.org/10.7554/eLife.83737.sa2

## Additional files

### Supplementary files
• Supplementary file 1. The epidemiological and clinical characteristics of the study groups.
• MDAR checklist

### Data availability
The data supporting the findings of this study are available within the paper and its supplementary information files. Source data are provided in this paper.

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
