## [Editor Report]

This important study shows that the expression of some inhibitory receptors on CD8 T cells is increased in people living with HIV (PLWH) and remains elevated even after years of viral suppression by antiretroviral therapy. The authors further provide convincing evidence that inhibition of TGIT partially restores the ability of CD8 T cells to produce CD107a but not the other functions and is relevant for researchers and clinicians interested in viral infections, especially HIV/AIDS.

---

## [Decision Letter]

**Decision letter after peer review:**

Thank you for submitting your article "Single-cell CD8^+^ dynamics uncover the reduction of CD107a TIGIT+ memory HIV-1-specific cells in PLWH over a decade of ART" for consideration by *eLife*. Your article has been reviewed by 3 peer reviewers, and the evaluation has been overseen by a Reviewing Editor and Murim Choi as the Senior Editor.

Essential revisions:

1. Revise the statistical analyses and consider correction for multiple comparisons and verify the validity of the use of mixing paired and unpaired comparisons in the same plots.

2. More primary data need to be included to prove the reliable detection of specific markers, such as Lag-3, Tim3, and CD39 and the potential biological of subtle differences should be critically discussed. Levels of baseline IR expression on naïve cells should be shown where appropriate.

3. The rationale for the selection of clusters for tSNE analysis needs to be provided.

4. Concerns about the utilization of a super-antigen as activators without knowing the absolute numbers of potential responding cells (Figure 3) and the interpretation of results shown in Figures 4 and 5 (see reviewer 3, points 4 and 5) need to be addressed.

5. Results from two data points do not allow us to conclude that there are continuous increases. Information on the magnitude of the HIV-specific CD8 T cell responses in the different cohorts should be provided.

6. It needs to be explained why the author directly focused on the CM component of CD8 HIV-specific responses instead of first looking at total HIV-specific CD8 T cells?

*Reviewer #1 (Recommendations for the authors):*

1. Please put this work in context with reference 25. That study demonstrated very similar results with respect to CD8 T cell expression of TIGIT; however, the prior study did show enhancement of IFN-g responses when TIGIT was blocked (CD107a was not analyzed).

2. For figure 1A, I believe there were 48 total PLWH analyzed not 24 as is currently shown (24 in S1 and 24 in S2). Also, the female/male ratio should be separated for both S1 and S2.

3. For the data demonstrated in figures 2-4, were the comparisons statistically corrected for multiple comparisons?

4. Please explain the importance of looking at SEB-stimulated T cells.

5. Please explain why only the three functional parameters were analyzed (CD107a, IFN-g, and IL-2).

6. It is unclear from the text, the effect of ART on these functional parameters. It appears that patients on ART for prolonged periods (S2) have some restoration of their HIV-specific CD8 T cells with respect to polyfunctionality. Please clarify as this is an important aspect with respect to the studies done for figure 5.

7. It appears that most of the analysis in figures 2-4 was unsupervised with the exception of supervised data shown for the HIV-specific data; however, it is unclear as to what parameters were supervised.

8. The following statement in the discussion "These data may support further investigations 309 on the potential use of TIGIT expression in CD8^+^ T cells as a biomarker of immune activation through residual replication in PLWH on ART" needs more justification. There are numerous publications showing that residual replication does not significantly occur and several demonstrate that continued immune activation is due to microbial translocation. The latter is supported by the current work whereby CD4 T cell counts correlate with TIGIT expression.

*Reviewer #2 (Recommendations for the authors):*

Statistics (all figures):

The statistical analyses should be revised, ideally with a statistician. Correction for multiple comparisons should be considered, and the validity of the use of mixing paired and unpaired comparisons in the same plots verified.

Figure 1: The levels of baseline expression of IRs on phenotypically naïve cells should be presented as well. While they are expected to be low, different cytokines can upregulate IRs on T cells in the absence of TCR signaling. Figure 1FandG: The authors focus on S2 in the text. They should consider also mentioning that they found significant negative correlations between CD4^+^ T-cell counts and the frequency of CM CD8^+^ T Cells expressing 2 and >1 IRs in the S1 condition.

Figure 2: Lines 198 -200 and at other places in the manuscript. The authors mention a continuous increase of effector-like clusters (and elsewhere, of other changes). It is not possible to confirm that these changes are a continuous process with only two time points, this statement should be revised.

Figure 3: The interpretation of the SEB-responsive cells seems to be "cellular clusters susceptible to TCR activation". However, SEB as a superantigen will stimulate only some Vbeta families. This should be clarified.

Figure 4: Some information should be given on the magnitude of the HIV-specific CD8 T cell responses in the different cohorts, and how this magnitude over time for S1 and S2 pairs. It is important, because it may change the interpretation of the shift in the relative proportions of the clusters observed (absolute attrition of some? Or the expansion of others?).

The authors directly focus on the CM component of these HIV-specific responses, it is unclear why and it would be important to look first at total HIV-specific CD8 T cells. If there is a shift in the memory differentiation pattern (e.g, in relative proportions of CM vs EM and TM over time), this may change the findings.

The authors stimulate with HIV-1 GAG to select for HIV-1 specific CD8^+^ T cells. Have they also tested other antigens (Nef, Env, Pol) in a subset of patients? Would they expect the results to be similar?

Figure 5: The functional assays with TIGIT blockade are limited and do not include other markers of cytotoxic cells (perforin, granzyme B expression…). It is not clear how these subsets compare to the other CD8 clusters in terms of CD107 expression.

Does the short-term ICB result in any changes to cell viability?

*Reviewer #3 (Recommendations for the authors):*

1. Data for Lag-3, Tim3, and CD39 shown in supplementary figure 1A does not appear to demonstrate reliable detection. Additional data should be shown to demonstrate convincing detection of these markers. Importantly, such raw data also needs to be shown for each of the stimulation conditions, and in the context of the functional outputs.

2. For the data shown in Figures 2,3,4, it is unclear why the stated number of clusters was chosen for the tSNE analysis. Whether this leads to the detection of meaningless clusters is unclear. In addition, in some cases, populations are grouped together, yet some of these grouped clusters appear disparate.

3. Many many statistical comparisons are made, yet there is no discussion of correction for multiple comparisons.

4. The differences between many groups appear very subtle despite being statistically different (pending adjustment for multiple comparisons). The authors should consider carefully what may be biologically relevant in the discussion.

5. The data analysis in Figure 3 is fundamentally flawed because the authors used super antigen as a 'polyclonal' activator. This is a great T cell activator but has to be interpreted carefully because every donor has an inherently different 'maximal' response based on the proportion of T cells bearing the appropriate TCR-BV to respond to SEB. This means that directly comparing total responding cells between groups is not particularly informative. Furthermore, without knowing the absolute number of potential responding cells (which was not measured here), it is not appropriate to interpret functional deficiencies within the population. Also, it is not correct to conclude polyclonal activation using SEB, because the clonality measure requires TCR assessment – not performed here. Within any given SEB-responding memory subset it is formally possible that a monoclonal activation could occur.

6. In Figures 4 and 5, it is difficult to interpret the data without knowing the actual magnitude of the responses to HIV, and the number of responding events recovered in any given subset examined. Did the authors have a cutoff for a minimum number of events to consider a positive response- both overall, and also within the subset populations?

7. Figure 5, the use of tSNE analysis does not seem necessary when memory subsets are simply examined with or without blockade. Also, how the memory subsets were defined should be described.

Stylistic comments.

1. The use of 'single-cell analysis' in the title and abstract (as well as several times in the paper) seems somewhat inappropriate at times given the more broad use of this term when referring to single-cell genomic studies. This manuscript is simply a flow cytometry study, which by definition is a single cell, but rarely described as such.

2. There are numerous stylistic and grammatical errors that should be fixed after careful reading; additionally:

– Lines 101-103 and 105-106 basically say the same thing.

– Lines 65-66, and 230-231 are not sentences.

– Line 205. define ICB.

– Line 267. fix 'PBCM.'

---

## [Author Response]

Essential revisions:1. Revise the statistical analyses and consider correction for multiple comparisons and verify the validity of the use of mixing paired and unpaired comparisons in the same plots.

According to the reviewers, we have revised the statistical analyses and included the correction for multiple comparisons, modifying the results across Figures and Supplemental Figures. Regarding the statistics, we changed Figures 2, 3, and 4 to have the intergroup and intragroup comparisons in separate graphs to clarify the information. In addition, we have included in the material and methods section and figure legends information regarding the statistical analyses and the correction for multiple comparisons (Holm’s method) performed when appropriate. Specific questions raised by the reviewers are included in the following sections.

2. More primary data need to be included to prove the reliable detection of specific markers, such as Lag-3, Tim3, and CD39 and the potential biological of subtle differences should be critically discussed. Levels of baseline IR expression on naïve cells should be shown where appropriate.

In agreement with the reviewers, we have provided primary data on reliable detection of LAG-3, TIM-3, and CD39 in Figure 1 —figure supplement 1, and data on baseline IR expression in naïve cells in Figure 1 and Figure 1 —figure supplement 2. Also, the potential biological implications of subtle and transient differences between IRs expression in CD8^+^ have been now included in the Discussion section (page 13).

Regarding the reliable detection of LAG-3, TIM-3, and CD39 markers. These markers have lower frequencies and fluorochrome intensities than PD-1 and TIGIT, comprising < 10% of total CD8^+^ T cells and subsets. To support the reliable detection, we have now provided source data of flowcytometry intensities for each fluorochrome compared with the Fluorescent minus one (FMO) intensity included now in Figure 1 —figure supplement 1. As shown in the figure, we can observe reliable detection for all the IRs markers despite differences in the intensities between fluorochromes compared with FMOs.

Regarding the baseline expression of IRs in naïve CD8^+^ T cells, we have now included data when appropriate, as suggested by reviewers. Information regarding the baseline expression of IR in naïve cells is now included in Figure 1 and Figure 1 —figure supplement 2. These data indicate very low or undetectable levels of baseline IR expression in naïve CD8^+^ T cells and subsets for all the IRs studied (TIGIT, PD-1, LAG-3, TIM-3, and CD39). This information further supports the reliable detection of IRs expression in total and CD8^+^ cellular subsets. Including the information on naïve CD8^+^ T cells brought new information regarding the composition of naïve populations by permutation analyses and demonstrated an interesting the reduction of CD8^+^ naïve cells expressing no IRs and the increase of naïve cells expressing one IR (Figure 1C).

3. The rationale for the selection of clusters for tSNE analysis needs to be provided.

In agreement with the reviewers, we now provided the rationale for cluster identification in the Material and Methods under the Unsupervised immunophenotype data analysis section (pages 20-21): “We discovered cell communities using the Phenograph clustering technique. It operates by computing the Jaccard coefficient between nearest neighbours, which was set to 30 in all executions, and then locating cell communities (or clusters) using the Louvain method. This creates a network indicating phenotypic similarities between cells. The netSNE maps included representations of the identified cell communities, and additionally, we built a heatmap with the clusters in the columns and the markers of interest in the rows to better comprehend the phenotypical interpretation of each cluster.”

4. Concerns about the utilization of a super-antigen as activators without knowing the absolute numbers of potential responding cells (Figure 3) and the interpretation of results shown in Figures 4 and 5 (see reviewer 3, points 4 and 5) need to be addressed.

We appreciate the reviewers' concerns about using superantigen Staphylococcal enterotoxin B (SEB) as an activator and the need for information about the absolute number of potential responding cells. Considering these comments, we have now included all the information required under de Results section. Unsupervised phenotypic characterisation of SEB-activated CD8^+^ T cells in PLWH in ART. In this section, we now justified the use of SEB to obtain complementary information on T-cell activation in response to pathogens involved in the disease by stimulating TCR-VB clonotypes, and previous studies in HIV-1 support the use of SEB to evaluate the effect of ICB in the recovery of T cell function (31,49,50). In addition, we provided information on the supervised analyses regarding the total frequency of SEB-activated CD8^+^ T cells presented in Figure 3 —figure supplement 1, revised the Results section accordingly, and included the lack of TCR sequencing as a limitation for data interpretation in the Discussion section.

5. Results from two data points do not allow us to conclude that there are continuous increases. Information on the magnitude of the HIV-specific CD8 T cell responses in the different cohorts should be provided.

We agree with the reviewers that the analyses of two time-points may not allow to conclude continuous increases. According with the comment, we have now rephrased the content of the manuscript when appropriate, including the title of the article, excluding the concept of dynamics. In agreement with the comment, we have now included all the information on the supervised analyses of the magnitude of total HIV-1-specific CD8 T cells in Figure 4 —figure supplement 1.

6. It needs to be explained why the author directly focused on the CM component of CD8 HIV-specific responses instead of first looking at total HIV-specific CD8 T cells?

According with the reviewers’ comments, we have now rephrased the Results section “Reduction of HIV-1-specific CD8^+^ T-cell clusters sharing memory-like phenotypes, TIGIT expression and low CD107a” include the analyses of all memory subsets, now in Figure 4F. We now have also included the supervised analyses of total HIV-1-specific CD8^+^ T cells (Figure 4 —figure supplement 1).

The focus on the memory compartment comes from identifying alterations of memory-like clusters by unsupervised analyses of HIV-1-specific CD8^+^ T cells and detecting continuous and significant increases of TIGIT CM cells in PLWH on ART (Figure 1D). This information is now summarised in the manuscript as follows (page 10): “These findings in memory-like clusters, together with the initial ones, accounting for increases in the frequency of TIGIT^+^ and TIGIT^+^TIM-3^+^ in CM and EFF cells on ART, led us to postulate then as potential markers of HIV-1-specific CD8^+^ T-cell dysfunction in PLWH on ART. Despite no significant changes observed in the total frequency of CD107a, IFNγ and IL-2 HIV-1-specific CD8^+^ T-cell responses between groups (Figure 4 —figure supplement 1A-B, Figure 4 – source data 3). The analyses of TIGIT and TIGIT+TIM-3 HIV-1-specific memory subsets revealed decreases in the frequency of CD107a TIGIT HIV-1-specific CD8^+^ T cells limited to the CM compartment (Figure 4F, Figure 4 – source data 4). No changes in IFNγ and IL-2 were observed. Furthermore, polyfunctional analysis of TIGIT HIV-1-specific CM CD8^+^s identified a decrease in monofunctional CD107a^+^ as well as in bifunctional CD107a^+^IFNγ^+^ and CD107^+^IL-2^+^ cells overtime on ART (Figure 4G). Overall, these data support a reduction of HIV-1-specific CD8^+^ T-cell clusters sharing memory-like phenotypes, TIGIT expression and low CD107a in PLWH on ART.”

Reviewer #1 (Recommendations for the authors):1. Please put this work in context with reference 25. That study demonstrated very similar results with respect to CD8 T cell expression of TIGIT; however, the prior study did show enhancement of IFN-g responses when TIGIT was blocked (CD107a was not analyzed).

According to this comment, we have now put in context our work with reference 25 in the Discussion section as follows (page 15): “These data contrast with Chew et al., which observed the recovery of IFNγ production by TIGIT blockade. However, they also reported a reduction in CD107a expression in TIGIT+ CD8 T cells in response to aCD3/aCD28 activation, supporting a dysfunctional profile of TIGIT+ CD8^+^ T cells. Differences between study groups accounting for time on ART, samples tested and interindividual variability of in vitro ICB experiments may account for some of the differences observed.”

2. For figure 1A, I believe there were 48 total PLWH analyzed not 24 as is currently shown (24 in S1 and 24 in S2). Also, the female/male ratio should be separated for both S1 and S2.

To clarify this point, we have accordingly revised Figure 1A, including subjects, samples, and female/male ratio, Supplementary Table 1 and the Study groups section in the manuscript. In brief, we analysed 48 PLWH, 24 in Early infection and 24 in ART. From those 24 in ART, we analysed 48 samples, 24 at the S1 and 24 at the S2 time points. The female/male ratio is the same for S1 and S2 samples as all samples come from the same 24 individuals selected in ART.

3. For the data demonstrated in figures 2-4, were the comparisons statistically corrected for multiple comparisons?

According to the reviewer, we have revised all the statistical analysis for the manuscript and corrected for multiple comparisons when appropriate using Holm's method. The information regarding the statistical method is included in the Materials and methods section under Statistics (page 20-21) and figure legends when appropriate.

4. Please explain the importance of looking at SEB-stimulated T cells.

Considering this comment, we have now included all the information required under de Results section, Unsupervised phenotypic characterisation of SEB-activated CD8^+^ T cells in PLWH in ART (page 8). Briefly, the use of Staphylococcal enterotoxin B (SEB) allows to characterise T-cell responses (in magnitude and function) to superantigen stimulation providing a complementary read out of response to pathogens involved in disease to antigen-specific T-cell responses. SEB has been previously used to monitor antigen-independent TCR activation through the cross-link of MHC class II and TCR VB of the T cell lymphocytes (Kou et al. 1998) and in vitro effect of the ICB (Lewin et al. 2022, and Teigler et al. 2017). We acknowledge the limitations of this kind of analysis in terms of TCR response and the functional profile observed, but also the interest of SEB-activated CD8^+^ T cells as complementary information to the antigenspecific responses to potentially detect functional defects.

5. Please explain why only the three functional parameters were analyzed (CD107a, IFN-g, and IL-2).

We prioritise the analyses of five IRs (TIGIT, PD-1, LAG-3, TIM-3 and CD39) and lineage markers (CD45RA, CCR7 and CD27) in our panel, adding relevant functional markers such as CD107a, IFNγ and IL-2 that provide general and complementary aspects of CD8^+^ T-cell functionality. From the functional parameters, we selected CD107a because it is widely used to detect cytolytic activity in CD8^+^ T cells as an early marker based on degranulation (Voskoboinik, et al. 2015, Aktas et al. 2009), we selected IFNγ because it is a classical moderator of cellmediated immunity with pro-inflammatory actions and enhances the antiviral effects of CD8^+^ T cells (Bhat et al. 2017) and selected IL-2 because it plays a crucial role in the expansion of CD8^+^ T cells, regulates proliferation and homeostasis, and long-term memory responses of T-cell responses (Gaffen and K. D. Liu 2004, Akdis et al. 2016). The relevant references to support using these three functional parameters have been included in the manuscript (references 51 to 55).

6. It is unclear from the text, the effect of ART on these functional parameters. It appears that patients on ART for prolonged periods (S2) have some restoration of their HIV-specific CD8 T cells with respect to polyfunctionality. Please clarify as this is an important aspect with respect to the studies done for figure 5.

According to the reviewer, we revised the functional data of HIV-1-specific CD8^+^ T cell responses. We have clarified the information in the Results section, new Figure 4, and new Figure 4 —figure supplement 1. All the information required is under de Results section, Reduction of HIV-1-specific CD8^+^ T-cell clusters sharing memory-like phenotypes, TIGIT expression and low CD107a (page 9). Briefly, Figure 4F included scatter plots showing median and interquartile ranges of CD107a, IFNγ, and IL-2 expression in TIGIT and TIGIT+TIM-3 HIV1-specific memory CD8^+^ T cell subsets. This analysis revealed decreased CD107a expression limited to the CM compartment in TIGIT HIV-1-specific CD8^+^ T cells. In addition, we have now included in Figure 4G the polyfunctional analyses of CD107a, IFNγ, and IL-2 expression for TIGIT HIV-1-specific CM cells. The new polyfunctional analysis identified monofunctional CD107a+ cells as the most common phenotype reduced in TIGIT HIV-1-specific CM cells, followed by bifunctional CD107a+IFNγ+ and CD107a+IL-2+ double-positive cells overtime on ART. These data do not support the restoration of polyfunctional responses over prolonged periods of ART (S2).

7. It appears that most of the analysis in figures 2-4 was unsupervised with the exception of supervised data shown for the HIV-specific data; however, it is unclear as to what parameters were supervised.

In agreement with the comment, we have now included whether we use supervised or unsupervised analyses in the text and figure legends for analyses corresponding to Figures 2-4. For SEB-activated CD8^+^ T cells, we used unsupervised net-SNE analyses and supervised classical analyses (Figure 3 and Figure 3 —figure supplement 1, respectively). Similarly, for HIV-1 -specific CD8^+^ T-cells, unsupervised net-SNE analyses and supervised classical analyses were combined across data sets (Figure 4 and Figure 4 —figure supplement 1, respectively).

8. The following statement in the discussion "These data may support further investigations 309 on the potential use of TIGIT expression in CD8^+^ T cells as a biomarker of immune activation through residual replication in PLWH on ART" needs more justification. There are numerous publications showing that residual replication does not significantly occur and several demonstrate that continued immune activation is due to microbial translocation. The latter is supported by the current work whereby CD4 T cell counts correlate with TIGIT expression.

According to the reviewer's comment, we have modified the content of the corresponding paragraph to clarify our statement focused on the association between TIGIT expression, poor immune recovery and persistent immune activation. We have now rewritten the content in the discussion as follows (page 12): “These data support continuous expression of TIGIT despite ART in agreement with previous studies (24,25,30,57,59) and uncover novel associations between TIGIT expression in CD8^+^ T cells and poorer immune status in PLWH on ART. Thus, these data indicate a specific contribution of TIGIT expression to persistent immune activation and poor CD4^+^ recovery on ART.”

Reviewer #2 (Recommendations for the authors):Statistics (all figures):The statistical analyses should be revised, ideally with a statistician. Correction for multiple comparisons should be considered, and the validity of the use of mixing paired and unpaired comparisons in the same plots verified.

According to the reviewer, we have revised all the statistical analysis to correct our findings for multiple comparisons when appropriate using Holm's method. The information regarding the statistical method is included in the statistics section of the Material and Methods and the figure legends when appropriate.

Figure 1: The levels of baseline expression of IRs on phenotypically naïve cells should be presented as well. While they are expected to be low, different cytokines can upregulate IRs on T cells in the absence of TCR signaling.

Data on baseline levels of IRs expression in naïve CD8^+^ T cells have been included in Figure 1 B-C and Figure 1 —figure supplement 2 B-C. These data demonstrate very low basal levels of IR expression in naïve CD8^+^ T cells across the IRs analysed (TIGIT, PD-1, LAG-3, TIM-3 and CD39). Including the information of basal IRs expression in naïve CD8^+^ T cells supports a reliable detection of IRs and uncovered new information of IRs expression now included in the Results section under, Alterations in CD8^+^ T-cell IRs frequencies and expression patterns in PLWH are not mitigated by ART (page 5).

Figure 1F and G: The authors focus on S2 in the text. They should consider also mentioning that they found significant negative correlations between CD4^+^ T-cell counts and the frequency of CM CD8^+^ T Cells expressing 2 and >1 IRs in the S1 condition.

According to the reviewer, we have clarified this point in the Results section under, Alterations in CD8^+^ T-cell IRs frequencies and expression patterns in PLWH are not mitigated by ART (page 6) as follows: “Focusing on S1 (Figure 1F), we found significant negative correlations between CD4^+^ T-cell counts and frequencies of CM CD8^+^ T cells expressing 2 (p=0.0054, r=0.64) and > 1 IRs (p=0.0087, r=-0.61). Focusing on S2 (Figure 1G), we observed significant negative correlations between CD4^+^ T-cell counts and the frequency of total CD8^+^ T cells expressing TIGIT^+^ (p=0.0157, r=-0.58), expressing 1 IRs (p=0.0386, r=-0.54) or >1 IRs (p=0.0386, r=-0.51). At the level of CD8^+^ T-cell subsets, the expression of 2 IRs in CM and >1 IR in TM negatively correlated with CD4^+^ T-cell counts (p=0.0072, r=-0.64; p=0.0346, r=-0.52, respectively) (Figure 1G)”.

Figure 2: Lines 198 -200 and at other places in the manuscript. The authors mention a continuous increase of effector-like clusters (and elsewhere, of other changes). It is not possible to confirm that these changes are a continuous process with only two time points, this statement should be revised.

In agreement with the comment, the statement has been revised across the manuscript. We agree with the reviewer that the analyses of two time points may not allow us to conclude a continuous increase of effector-like clusters. Accordingly, we have rephrased the manuscript when appropriate.

Figure 3: The interpretation of the SEB-responsive cells seems to be "cellular clusters susceptible to TCR activation". However, SEB as a superantigen will stimulate only some Vbeta families. This should be clarified.

In agreement with the comment, we have clarified the information and removed the term TCR activation by SEB-activated CD8^+^ T cells. We have included a paragraph under the Results section, Unsupervised phenotypic characterisation of SEB-activated CD8^+^ T cells in PLWH in ART (page 8) as follows: “Then, we evaluate CD8^+^ T-cell responses by bacterial superantigen activation with Staphylococcal enterotoxin B (SEB). Using SEB can provide complementary information on T-cell activation in response to pathogens involved in the disease by stimulating TCR-VB clonotypes (31,49,50).”

Figure 4: Some information should be given on the magnitude of the HIV-specific CD8 T cell responses in the different cohorts, and how this magnitude over time for S1 and S2 pairs. It is important, because it may change the interpretation of the shift in the relative proportions of the clusters observed (absolute attrition of some? Or the expansion of others?).The authors directly focus on the CM component of these HIV-specific responses, it is unclear why and it would be important to look first at total HIV-specific CD8 T cells. If there is a shift in the memory differentiation pattern (e.g, in relative proportions of CM vs EM and TM over time), this may change the findings.

According to the reviewer, we have revised the information on total HIV-1-specific CD8^+^ T-cell responses and evaluated the magnitude of the response base on CD107a, IFNγ and IL-2 expression between study groups (Figure 4 —figure supplement 1). No differences in total HIV1-specific responses were observed between groups and over time on ART (S1 vs S2). Based on the unsupervised analyses, we focused on the memory compartment and TIGIT and TIGIT+TIM3 HIV-1-specific CD8^+^ T cells and found differences by classical supervised analysis on the CM compartment (Figure 4F). In addition, we characterise the TIGIT CM HIV-1-specific CD8^+^ T cells in terms of CD107a expression and polyfunctional profile with IFNγ and IL-2 (Figure 4G). We have clarified the information in the Results section, new Figure 4, and new Figure 4 —figure supplement 1. All the information required is under de Results section, Reduction of HIV-1-specific CD8^+^ T-cell clusters sharing memory-like phenotypes, TIGIT expression and low CD107a (page 9-10).

The authors stimulate with HIV-1 GAG to select for HIV-1 specific CD8^+^ T cells. Have they also tested other antigens (Nef, Env, Pol) in a subset of patients? Would they expect the results to be similar?

We acknowledge the interest in testing additional antigens to select for HIV-1-specific CD8^+^ T cells. We have not performed in this study stimulations with other antigens, including Nef, Env, and Pol, in a subset of patients. We acknowledge this point as study limitation and the potential interest of testing complementary antigens to select HIV-1-specific CD8^+^ T cells against early (Nef) and late virally-expressed proteins (Gag, Env). Indeed, previous studies by our group (Kloverpis et al., 2013, Ruiz et al. 2019) support the differences in the early vs late antigen recognition of HIV-1 proteins in the efficacy and kinetics of killing infected cells by CD8^+^ T cells. We have included this information in the Discussion section (page 15): “We acknowledge several study limitations; First, the sample size of study groups and the use of peripheral blood samples underestimate the potential contribution of TIGIT expression to Tex in lymphoid tissues. Second, the use of only Gag as stimuli for the characterization of HV-1-specific CD8^+^ T cell responses in the absence of TCR sequencing. Using alternative HIV-1 antigens such as Nef, Env or Pol may provide additional information on the profile of CD8^+^ T-cell functional responses against early and late-expressed viral proteins in PLWH on ART (68, 69).

Figure 5: The functional assays with TIGIT blockade are limited and do not include other markers of cytotoxic cells (perforin, granzyme B expression…). It is not clear how these subsets compare to the other CD8 clusters in terms of CD107 expression.Does the short-term ICB result in any changes to cell viability?

We acknowledge the interest in testing additional markers, including perforin and granzyme B, for a more complete characterisation of the recovery of the cytotoxic potential of HIV-1-specific CD8^+^ T cells. We have now acknowledged this point as a study limitation and included this information in the Discussion section (page 15): “Third, limited ICB experiments to CD107a, INFγ and IL-2 functional markers without complementary cytotoxic markers (perforin, granzyme B). Forth, complementary transcriptomic, epigenetic, and metabolic markers are needed for a complete description of Tex's immune signatures linked to TIGIT expression in HIV-1 specific CD8^+^ T cells in PLWH on ART”.

Additionally, we have revised the information regarding changes in cell viability by short-term ICB. We have based our analysis on the frequency of total live lymphocytes using live/dead probe by flow cytometry and compared the viability of the PBMCs in the presence of HIV-1 (HIV-1 Gag peptide pool) and in the presence of HIV-1+ IgG isotypes alone or combined. A similar comparison was performed between PBMCs in the presence of HIV-1 and the presence of blockade antibodies aTIGIT and aTIM-3 alone or combined. We have performed paired analysis (Wilcoxon matched-pairs signed rank test) correcting for multiple comparisons (Holm´s method) between conditions. Our results indicated no changes in cellular viability by short-term ICB as represented in Author response image 1:

**Author response image 1. sa2fig1:** Graphs represent paired comparisons of the frequency of live cells in PBMC samples in short-term ICB studies. A. Graphs indicate the percentage of live cells in HIV-1 condition (Gag peptide pool) compared to HIV-1 + IgG2 isotype, HIV-1 + IgG1 isotype and HIV-1 + IgG2a+IgG1. B. Graphs indicate the % of live cells in HIV-1 compared to HIV-1 + aTIGIT, HIV-1 + aTIM-3 and HIV-1 + aTIGIT+ aTIM-3.

Reviewer #3 (Recommendations for the authors):1. Data for Lag-3, Tim3, and CD39 shown in supplementary figure 1A does not appear to demonstrate reliable detection. Additional data should be shown to demonstrate convincing detection of these markers. Importantly, such raw data also needs to be shown for each of the stimulation conditions, and in the context of the functional outputs.

In agreement with the reviewer, we have demonstrated reliable detection of IRs in two ways; we have added Figure 1 —figure supplement 1B, including the comparison for the staining of all markers (TIGIT, PD-1, LAG-3, TIM-3 and CD39) in CD8^+^ T cells with the corresponding FMO in cryopreserved PBMCs. This information supports the reliable detection of all the markers data is similar for the SEB or HIV-1 condition. Also, including the information on basal IRs expression in naïve CD8^+^ T cells in Figure 1 and Figure 1 —figure supplement 2, requested by the previous reviewer, provides further evidence of the reliable detection of these markers.

2. For the data shown in Figures 2,3,4, it is unclear why the stated number of clusters was chosen for the tSNE analysis. Whether this leads to the detection of meaningless clusters is unclear. In addition, in some cases, populations are grouped together, yet some of these grouped clusters appear disparate.

We have revised and clarified the information regarding the tSNE analyses and eliminated cluster aggrupation to avoid problems in the interpretation of data analysis and representation. We now provided the rationale for cluster identification in the Material and Methods under the Unsupervised immunophenotype data analysis section (pages 19-20): “We discovered cell communities using the Phenograph clustering technique. It operates by computing the Jaccard coefficient between nearest neighbours, which was set to 30 in all executions, and then locating cell communities (or clusters) using the Louvain method. This creates a network indicating phenotypic similarities between cells. The netSNE maps included representations of the identified cell communities, and additionally, we built a heatmap with the clusters in the columns and the markers of interest in the rows to better comprehend the phenotypical interpretation of each cluster.”

3. Many many statistical comparisons are made, yet there is no discussion of correction for multiple comparisons.

We have revised the statistical analyses and included the correction for multiple comparisons, modifying the results across Figures and Supplemental Figures. Regarding the statistics, we changed Figures 2, 3, and 4 to have the intergroup and intragroup comparisons in separate graphs to clarify the information and correct for multiple comparisons. We have included information in the material and methods section and Figure legends regarding the statistical analyses and the correction for multiple comparisons (Holm’s method) performed when appropriate.

4. The differences between many groups appear very subtle despite being statistically different (pending adjustment for multiple comparisons). The authors should consider carefully what may be biologically relevant in the discussion

After adjusting for multiple comparisons, we revised the manuscript's analyses, figures and corresponding text. Additionally, we have carefully revised the discussion pointing to the potential biological significance of our findings.

5. The data analysis in Figure 3 is fundamentally flawed because the authors used super antigen as a 'polyclonal' activator. This is a great T cell activator but has to be interpreted carefully because every donor has an inherently different 'maximal' response based on the proportion of T cells bearing the appropriate TCR-BV to respond to SEB. This means that directly comparing total responding cells between groups is not particularly informative. Furthermore, without knowing the absolute number of potential responding cells (which was not measured here), it is not appropriate to interpret functional deficiencies within the population. Also, it is not correct to conclude polyclonal activation using SEB, because the clonality measure requires TCR assessment – not performed here. Within any given SEB-responding memory subset it is formally possible that a monoclonal activation could occur.

We appreciate the reviewer concerns about using superantigen Staphylococcal enterotoxin B (SEB) as an activator and the need for information about the absolute number of potential responding cells. Considering these comments, we have now included all the information required under de Results section, Unsupervised phenotypic characterization of SEB-activated CD8^+^ T cells in PLWH in ART and removed the concept of polyclonal activator from the manuscript. In this section, we now justified the use of SEB to obtain complementary information on T-cell activation in response to pathogens involved in the disease by stimulating TCR-VB clonotypes, and previous studies in HIV-1 support the use of SEB to evaluate the effect of ICB in the recovery of T cell function (31,49,50). In addition, we provided information on the supervised analyses regarding the total frequency of SEB-activated CD8^+^ T cells presented in Figure 3 —figure supplement 1, and revised the Results section carefully according to the findings. Also, we included the lack of TCR assessment by sequencing as a limitation for data interpretation in the Discussion section.

6. In Figures 4 and 5, it is difficult to interpret the data without knowing the actual magnitude of the responses to HIV, and the number of responding events recovered in any given subset examined. Did the authors have a cutoff for a minimum number of events to consider a positive response- both overall, and also within the subset populations?

We appreciate the reviewer's concerns regarding Figures 4 and 5. For Figure 4, we have analysed the information on total HIV-1-specific CD8^+^ T-cell responses and evaluated the magnitude of the response base on CD107a, IFN and IL-2 expression between study groups by supervised analyses (Figure 4 —figure supplement 1). Also, the information for the cut-off values and minimum numbers of events to consider a positive CD8^+^ T cell functional response based on CD107a, IFN and IL-2 by flow cytometry analyses is now included in the Material and Methods section (page 19) as follows: “We performed two technical replicates for SEB-activated and HIV1-specific CD8^+^ T-cell cytokine production. We considered the cytokine response positive after background subtraction (mean of two technical replicates) used as the cut-off value. For each independent sample, we recorded a median of 1,000 events and 50 events positive for cytokines for total and CD8^+^ T cell subsets, respectively.”

7. Figure 5, the use of tSNE analysis does not seem necessary when memory subsets are simply examined with or without blockade. Also, how the memory subsets were defined should be described.

In agreement with the comment, we have now modified Figure 5 and include representative dot plots of the ICB experiments for the different conditions tested (basal, Isotypes, αTIGIT, αTIM3, and αTIGIT+αTIM-3 blockade) and functional markers monitored (CD107a, IFNγ and IL-2) (Figure 5A). Also, we included specific net-SNE projections to represent the extent of mAb blockade (Figure 5B).

The definition of memory subsets CD8^+^ T cells and the gating strategy followed for flow cytometry analysis is included in Figure 1 —figure supplement 1 and the Materials and methods section.

Stylistic comments.1. The use of 'single-cell analysis' in the title and abstract (as well as several times in the paper) seems somewhat inappropriate at times given the more broad use of this term when referring to single-cell genomic studies. This manuscript is simply a flow cytometry study, which by definition is a single cell, but rarely described as such.

According to the comment, we have rephrased the manuscript's content when appropriate, including the title and excluding the concept of single-cell analysis across the manuscript.

2. There are numerous stylistic and grammatical errors that should be fixed after careful reading; additionally:

According to the comment, we have revised the complete manuscript for stylistic and grammatical errors

– Lines 101-103 and 105-106 basically say the same thing.

Revised.

– Lines 65-66, and 230-231 are not sentences.

Revised.

– Line 205. define ICB.

Revised.

– Line 267. fix 'PBCM.'

Revised.